# Stable Minima Cannot Overfit in Univariate ReLU Networks: Generalization by Large Step Sizes

**Dan Qiao**
CSE, UC San Diego
d2qiao@ucsd.edu

**Kaiqi Zhang**
CS, UC Santa Barbara
kzhang70@ucsb.edu

**Esha Singh**
CSE, UC San Diego
e3singh@ucsd.edu

**Daniel Soudry**
Technion – Israel Institute of Technology
daniel.soudry@gmail.com

**Yu-Xiang Wang**
Halıcıoğlu Data Science Institute, UC San Diego
yuxiangw@ucsd.edu

## Abstract

We study the generalization of two-layer ReLU neural networks in a univariate nonparametric regression problem with noisy labels. This is a problem where kernels (*e.g.* NTK) are provably sub-optimal and benign overfitting does not happen, thus disqualifying existing theory for interpolating (0-loss, global optimal) solutions. We present a new theory of generalization for local minima that gradient descent with a constant learning rate can *stably* converge to. We show that gradient descent with a fixed learning rate $\eta$ can only find local minima that represent smooth functions with a certain weighted *first order total variation* bounded by $1/\eta - 1/2 + \widetilde{O}(\sigma + \sqrt{\text{MSE}})$ where $\sigma$ is the label noise level, $\text{MSE}$ is short for mean squared error against the ground truth, and $\widetilde{O}(\cdot)$ hides a logarithmic factor. Under mild assumptions, we also prove a nearly-optimal MSE bound of $\widetilde{O}(n^{-4/5})$ within the strict interior of the support of the $n$ data points. Our theoretical results are validated by extensive simulation that demonstrates large learning rate training induces sparse linear spline fits. To the best of our knowledge, we are the first to obtain generalization bound via minima stability in the non-interpolation case and the first to show ReLU NNs without regularization can achieve near-optimal rates in nonparametric regression.

## 1 Introduction

How do gradient descent-trained neural networks work? It is an intriguing question that depends on model architecture, data distribution, and optimization algorithms used for training [Zhang et al., 2021]. Specifically, in the overparameterized regime with specific random initialization of the weights, it was shown that gradient descent finds global optimal (0-loss or interpolating) solutions despite the non-convex objective function [Jacot et al., 2018, Du et al., 2018, Liu et al., 2022]. It was also shown that among the (many) global optimal solutions, the particular solutions that are selected by gradient descent often do not overfit despite having 0 training error [Chizat et al., 2019, Arora et al., 2019, Mei et al., 2019], sometimes even if the data is noisy — a phenomenon known as "benign overfitting" [e.g., Belkin et al., 2019, Bartlett et al., 2020, Frei et al., 2022].

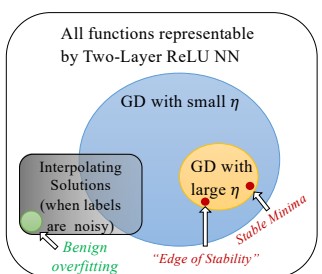

Figure 1: We show that "Large step size selects simple functions that generalize."

38th Conference on Neural Information Processing Systems (NeurIPS 2024).

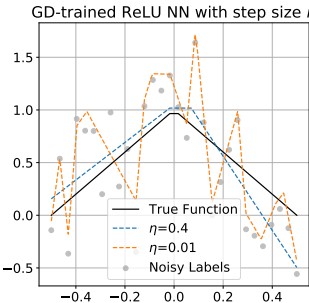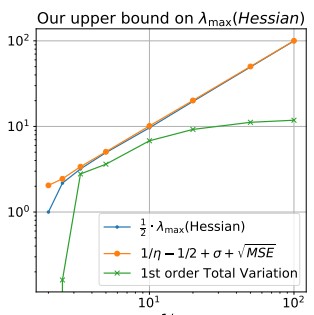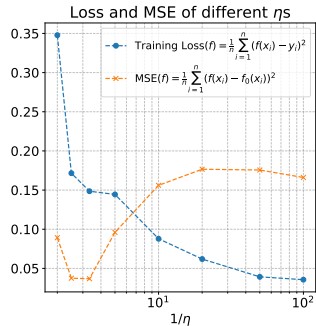

Figure 2: Empirical evidence of our claim. Constant step size gradient descent-trained two-layer ReLU neural networks generalize because of minima stability. The **left panel** shows that with increasing step size, gradient descent finds smoother solutions (linear splines) with a smaller number of knots. The **middle panel** illustrates our theoretical result with a *numerically accurate* upper bound using $1/\eta + O(1)$ of the curvature and TV1-complexity of the smooth solution. The **right panel** shows that tuning $\eta$ gives the classical U-shape bias-variance tradeoff for overparameterized NN.

What is less well-known is that interpolating solutions do overfit for ReLU neural networks (ranging from tempered to catastrophic) [Mallinar et al., 2022, Joshi et al., 2023, Haas et al., 2023] and the generalization bounds in the kernel regime are provably suboptimal for certain univariate nonparametric regression problems [Suzuki, 2018, Zhang and Wang, 2022]. These "exceptions" significantly limit the abilities of the kernel theory or "benign overfitting" theory in predicting the performance of an overparameterized neural network in practice.

In many learning problems with noisy data, the best solutions are simply not among those that interpolate the data. For example, no interpolating solutions can be *consistent* in a *fixed-design* nonparametric regression problem. Even if interpolating solutions that satisfy benign overfitting can be found, they could be undesirable due to their "spikiness" [Haas et al., 2023] and lack of robustness [Hao and Zhang, 2024]. In addition, it was reported that when the label is noisy, it takes much longer for gradient descent to overfit [Zhang et al., 2021]. Most practical NN training would have entered the *Edge-of-Stability* regime [Cohen et al., 2020] or stopped before the interpolation regime kicks in.

These observations motivate us to come up with an alternative theory for gradient-descent training of overparameterized neural networks that do not require interpolation.

## 1.1 Summary of Contributions

In this paper, we present a new theory of generalization for solutions that gradient descent (GD) with a fixed learning rate can *stably* converge to. Specifically:

1. We show that for 1D nonparametric regression ($n$ data points with noisy labels), the solutions that GD can stably converge to *must be* regular functions with small (weighted) first-order *total variation* (Theorem 4.1 and Corollary 4.2), thus promoting sparsity in the number of linear pieces. This generalizes the result of Mulayoff et al. [2021] by removing the "interpolation" assumption. We also show that in the noisy case, there is no "flat" interpolating solution and gradient descent cannot converge to them unless the learning rate is $O(1/n^2)$ no matter how overparameterized the two-layer ReLU network is (Theorem 3.1).

2. We show that such solutions (stable local minima that GD converges to) *cannot overfit*, in the sense that the *generalization gap* vanishes as $n \to \infty$ inside the strict interior of the data support (Theorem 4.3). Moreover, under a mild additional assumption on gradient descent finding solutions with training loss smaller than $\sigma^2$, we prove that these solutions achieve *near-optimal* rate for estimating (the strict interior of) *first-order bounded variation functions* (Theorem 4.4) — provably faster than any kernel ridge regression estimators, including neural networks in the "kernel" regime.

3. We conduct extensive numerical experiments to demonstrate our theoretical predictions, validate our technical assumptions, and illustrate the functional form of the ReLU NNs as well as the

learned basis functions that gradient descent finds with different step sizes. These results reveal new insights into how gradient descent training aggressively learns representation and induces implicit sparsity.

To the best of our knowledge, these results are new to this paper. We emphasize that (1) the training objective function is not explicitly regularized; (2) we do not early-stop training in favor of algorithmic-stability; and (3) the solutions that gradient descent converges to are not global optimal (interpolating) solutions unless the label noise is $0$. Our approach is new in that we directly analyze the complexity of a superset of solutions that gradient descent can stably converge to, which enables us to prove end-to-end generalization bounds that are near-optimal in nonparametric regression tasks.

Our analysis for gradient descent-training is categorically different from those in the "kernel" (a.k.a. "lazy") regime about interpolating solutions. Instead, we rigorously prove (and empirically demonstrate) that large step-size gradient descent do not behave this way and it does not converge to interpolating solutions. Our results fall into the non-kernel regime of neural network learning known as the "rich" (a.k.a. the "feature learning" or "sparse") regime [Chizat et al., 2019, Woodworth et al., 2020], in which the weights and biases can move arbitrarily far away from their initialization.

**Technical novelty.** The main technical innovation in our analysis is in handling the $\frac{1}{n}\sum_i(y_i - f_\theta(x_i))\nabla^2_\theta f_\theta(x_i)$ term that arises in the minima stability analysis when it was previously handled by Mulayoff et al. [2021] using interpolation, i.e., $y_i = f(x_i) \,\forall\, i \in [n]$. It turns out in the noisy-label case, we can decompose the term into a certain Gaussian complexity measure and a self-bounding style MSE of that $f_\theta$. A non-trivial step is to bound the largest eigenvalue of $\nabla^2 f_\theta(\cdot)$ by a constant which results in a uniform bound of both the Gaussian complexity and the MSE. These bounds themselves are vacuous in terms of the implied generalization error, but plugging them into the function-space constraint imposed by the noisy minima stability bound restricts the learned ReLU NN to be inside a weighted TV1 function class (Details in Theorem 4.1). This, in turn, allows us to *amplify* a vacuous MSE bound into a new MSE bound that is nearly optimal in the strict interior of the data support. The main technique in the last step involves bounding the metric entropy of the weighted TV1 class and then carefully working out a self-bounding (square loss) version of the Dudley's chaining argument [Wainwright, 2019].

**Disclaimers and limitations.** It is important to note that while we analyze gradient descent training of overparameterized neural networks, the computational claim is very different from those in the kernel regime. The analysis in the kernel regime ensures that gradient descent finds interpolating solutions efficiently. We do not have a comparable efficiency claim. While computational guarantees on stationary point (and local minima) convergence in non-convex optimization problems are well-understood [Ghadimi and Lan, 2013, Jin et al., 2017], we do not have guarantees on whether the solution that gradient descent finds satisfies our assumption on the training loss being "optimized" (smaller than label noise $\sigma^2$). Instead, our results provide a generalization gap bound for any stable solutions (Theorem 4.3) and a near-optimal excess risk (MSE vs ground truth) bound (Theorem 4.4) when the solution that GD finds happens to satisfy the assumption (empirically it does!). This is a meaningful middle ground between classical learning theory which does not concern optimization at all and modern theory that is fully optimization-dependent.

While we focus on (full batch) gradient descent for a clean presentation, the minima stability for stochastic gradient descent is immediate under the stochastic definition of minima stability [Mulayoff et al., 2021]. The same reason applies to us focusing on univariate nonparametric regression. Our technique can be used to generalize the multivariate function space interpretation of minima stability from Nacson et al. [2022] to the noisy case, but it will take substantial effort to formalize the corresponding generalization bounds in the multivariate case, which we leave as a future work.

## 1.2 Related Work and Implications of Our Results

**Generalization in Overparameterized NNs, Interpolation, and Benign Overfitting.** Most existing theoretical work on understanding the generalization of overparameterized neural networks focuses on the interpolation regime [Cao and Gu, 2019, Frei et al., 2022, Kou et al., 2023, Buzaglo et al., 2024]. Handling label noise either requires explicit regularization [Hu et al., 2020, Zhang and Wang, 2022], algorithmic stability through early stopping [Hardt et al., 2016, Richards and Kuzborskij, 2021], or carefully crafted data distribution that leads to a phenomenon known as benign overfitting [Bartlett et al., 2020, Frei et al., 2022, Kou et al., 2023]. Benign overfitting could happen for nonparametric

regression tasks [Belkin et al., 2019], but there is well-documented empirical and theoretical evidence that benign overfitting does not occur for regression tasks with ReLU activation [Mallinar et al., 2022, Haas et al., 2023, Joshi et al., 2023] and that the excess risk is required to be proportional to the standard deviation of the label noise [Kornowski et al., 2024]. We are the first to go beyond the interpolation regime and show that gradient descent-trained neural networks generalize in noisy regression tasks without explicit regularization.

**Implicit bias of gradient descent.** The implicit bias of gradient descent training of overparameterized NN is well-studied. It was shown that among the many globally optimal (interpolating) solutions, gradient descent finds the ones with the smallest norm in certain Hilbert spaces [Arora et al., 2019, Mei et al., 2019], classifiers with largest-margin [Chizat and Bach, 2020], or the smoothest cubic spline interpolation [Jin and Montúfar, 2023]. None of these results, however, imply generalization bounds when the labels are noisy. Interestingly, our results show that gradient descent with a large step size induces an implicit bias that resembles sparse L1-regularization rather than the dense L2 regularization from gradient flow [Jin and Montúfar, 2023].

**Implicit bias of minima stability.** The closest to our work is the line of work on the implicit bias of minima stability [Ma and Ying, 2021, Mulayoff et al., 2021, Wu and Su, 2023]. We build directly on top of the function-space interpretation of minima stability established by Mulayoff et al. [2021]. However, these works critically rely on the minima interpolating the data, which makes their results inapplicable to settings with label noise. Mulayoff et al. [2021] also did not establish formal generalization bounds. Ma and Ying [2021], Wu and Su [2023] do have generalization bounds, but (again) their results require interpolation and thus do not apply to our settings.

**Flat/Sharp Minima and generalization.** Our work is also connected to the body of work on the hypothesis that "flat local minima generalize better". Despite compelling empirical evidence [Hochreiter and Schmidhuber, 1997, Keskar et al., 2017], rigorous theoretical understanding of this hypothesis is still lacking [see, e.g., Wu and Su, 2023, and the references therein]. Our work contributes to this literature by formally proving that the hypothesis is real for two-layer ReLU NNs in a noisy regression task.

**Edge-of-Stability and Catapults.** Empirical observations on how large learning rate training of NN finds solutions with Hessian's largest eigenvalue dancing around $2/\eta$, i.e., "edge of stability" regime [Cohen et al., 2020]; and that the loss may go up first before going down to a good solution ("catapult") [Lewkowycz et al., 2020]. Existing theoretical understanding of these curious behaviors of GD training is still limited to toy-scale settings (e.g., Arora et al. [2022], Ahn et al. [2023], Kreisler et al. [2023]). Our work is complementary in that we provide generalization bound to the final solution GD stabilizes on no matter how GD gets there. Outside the context of GD and neural networks, "edge of stability" and the implicit bias of large step-size were observed for forward stagewise regression [see, e.g. Tibshirani, 2015, 2014, Section 4.4 and Page 42] albeit only empirically. Our results may provide a theoretical handle in formally analyzing these observations.

**Optimal rates of NNs in nonparametric regression.** Finally, it was previously known that neural networks can achieve optimal rates for estimating TV1 functions [Suzuki, 2018, Liu et al., 2021, Parhi and Nowak, 2021, Zhang and Wang, 2022]. Specifically, Savarese et al. [2019], Ongie et al. [2020], Parhi and Nowak [2021], Zhang and Wang [2022] show that weight decay in ReLU networks is connected to total variation regularization. However, these works assume one can solve an appropriately constrained or regularized empirical risk minimization problem. Our work is the first to show that the optimal rate is achievable with gradient descent without weight decay. In fact, it was a pleasant surprise to us that both weight decay and large learning rate induce total variation-like implicit regularization in the function space.

## 2 Notations and Problem Setup

Let us set up the problem formally. Throughout the paper, we use $O(\cdot), \Omega(\cdot)$ to absorb constants while $\widetilde{O}(\cdot)$ suppresses logarithmic factors. Meanwhile, $[n] = \{1, 2, \cdots, n\}$.

**Two-layer neural network.** We consider two-layer (*i.e.* one-hidden-layer) univariate ReLU networks,

$$\mathcal{F} = \left\{ f : \mathbb{R} \to \mathbb{R} \mid f(x) = \sum_{i=1}^{k} w_i^{(2)} \phi \left( w_i^{(1)} x + b_i^{(1)} \right) + b^{(2)} \right\}, \tag{1}$$

where the network consists of $k$ hidden neurons and $\phi(\cdot)$ denotes the ReLU activation function.

**Training data and loss function.** The training dataset is denoted by $\mathcal{D} = \{(x_i, y_i) \in \mathbb{R} \times \mathbb{R}, i \in [n]\}$. $\{x_i\}_{i=1}^n$ is assumed to be supported by $[-x_{\max}, x_{\max}]$ for some constant $x_{\max} > 0$. We focus on regression problems with square loss $\ell(f, (x, y)) = \frac{1}{2}(f(x) - y)^2$. The training loss is defined as $\mathcal{L}(f) = \frac{1}{2n} \sum_{i=1}^n (f(x_i) - y_i)^2$. Notice that $f$ is parameterized by $\theta := [w_{1:k}^{(1)}, b_{1:k}^{(1)}, w_{1:k}^{(2)}, b^{(2)}] \in \mathbb{R}^{3k+1}$. As a short hand, we define $\ell_i(\theta) := \ell(f_\theta, (x_i, y_i))$ and $\mathcal{L}(\theta) := \frac{1}{n} \sum_{i \in [n]} \ell_i(\theta)$.

**Gradient descent.** We focus on the Gradient descent (GD) learner, which iteratively updates $\theta$:

$$\theta_{t+1} = \theta_t - \eta \nabla \mathcal{L}(\theta_t), \ t \geq 0, \tag{2}$$

where $\eta > 0$ is the step size (a.k.a. learning rate) and $\theta_0$ is the initial parameter. Detailed calculation of gradient for two-layer ReLU networks is deferred to Appendix E. Below we define stability for local minima and discuss the conditions for a minimum to be stable.

**Twice differentiable stable local minima.** Similar to Mulayoff et al. [2021], we consider twice differentiable minima. According to Taylor's expansion around a twice differentiable minimum $\theta^\star$,

$$\mathcal{L}(\theta) \approx \mathcal{L}(\theta^\star) + (\theta - \theta^\star)^T \nabla \mathcal{L}(\theta^\star) + \frac{1}{2}(\theta - \theta^\star)^T \nabla^2 \mathcal{L}(\theta^\star)(\theta - \theta^\star), \tag{3}$$

where $\nabla^2 \mathcal{L}$ denotes the Hessian matrix and $\nabla \mathcal{L}(\theta^\star) = 0$. Therefore, as $\theta_t$ gets close to $\theta^\star$, the update rule for GD (2) can be approximated as $\theta_{t+1} \approx \theta_t - \eta \left( \nabla \mathcal{L}(\theta^\star) + \nabla^2 \mathcal{L}(\theta^\star)(\theta_t - \theta^\star) \right)$. Such approximation motivates the definition of linear stability, which is first stated in Wu et al. [2018].

**Definition 2.1** (Linear stability). With the update rule $\theta_{t+1} = \theta_t - \eta \left( \nabla \mathcal{L}(\theta^\star) + \nabla^2 \mathcal{L}(\theta^\star)(\theta_t - \theta^\star) \right)$, a twice differentiable local minimum $\theta^\star$ of $\mathcal{L}$ is said to be $\epsilon$ linearly stable if for any $\theta_0$ in the $\epsilon$-ball $\mathcal{B}_\epsilon(\theta^\star)$, it holds that $\limsup_{t \to \infty} \|\theta_t - \theta^\star\| \leq \epsilon$.

Note that different from previous works [Wu et al., 2018, Mulayoff et al., 2021], we remove the expectation before $\|\theta_t - \theta^\star\|$ since under GD everything is deterministic. Intuitively speaking, linear stability requires that once we have arrived at a distance of $\epsilon$ from $\theta^\star$, we end up staying in the $\epsilon$-ball $\mathcal{B}_\epsilon(\theta^\star)$. It is known that linear stability is connected to the flatness of the local minima.

**Lemma 2.2.** *Consider the update rule in Definition 2.1, for any $\epsilon > 0$, a local minimum $\theta^\star$ is an $\epsilon$ linearly stable minimum of $\mathcal{L}$ if and only if $\lambda_{\max}(\nabla^2 \mathcal{L}(\theta^\star)) \leq \frac{2}{\eta}$.*

The implication is that the set of stable minima is equivalent to the set of flat local minima whose largest eigenvalue of Hessian is smaller than $2/\eta$. The proof is adapted from Mulayoff et al. [2021] and we state the proof in Appendix C for completeness. When the result does not depend on $\epsilon$ (as above), we simply say "linearly stable". Throughout the paper, we overload the notation by calling a function $f = f_\theta$ linearly stable function if $\theta$ is linearly stable.

**"Edge of Stability" regime.** Extensive empirical and theoretical evidence (Cohen et al. [2020], Damian et al. [2024], and see Section 1.2) have shown that the threshold of linear stability (from Lemma 2.2) is quite significant in GD dynamics: GD iterations initially tend to exhibit "progressive sharpening", where $\lambda_{\max}(\nabla^2 \mathcal{L}(\theta_t))$ is increasing, until finally GD reaches the "Edge of Stability", where $\lambda_{\max}(\nabla^2 \mathcal{L}(\theta_t)) \approx 2/\eta$. We capture this phenomenon with the following definition.

**Definition 2.3** (Below Edge of Stability). We say that a sequence of parameters $\{\theta_t\}_{t=1,2,\dots}$ generated by gradient descent with step-size $\eta$ is $\epsilon$-approximately Below-Edge-of-Stability (BEoS) for $\epsilon > 0$ if there exists $t^* > 0$ such that $\lambda_{\max}(\nabla^2 \mathcal{L}(\theta_t)) \leq \frac{2e^\epsilon}{\eta}$ for all $t \geq t^*$. Any $\theta_t$ with $t \geq t^*$ is referred to as an $\epsilon$-BEoS solution.

The BEoS regime provides a strong justification for the connection between the step size $\eta$ and the largest eigenvalue of the Hessian matrix. It holds for all twice-differentiable solutions GD finds along the way — even if the GD does not converge to a (local or global) minimum. Empirically, BEoS is valid for both the "progressive sharpening" phase and the oscillating EoS phase for a small constant $\epsilon$.

Our goal in this paper is to understand generalization for both (twice differentiable) stable local minima (Definition 2.1) and any other solutions satisfying $\epsilon$-BEoS (Definition 2.3), which are both subsets of

$$\mathcal{F}(\eta, \epsilon, \mathcal{D}) := \left\{ f_\theta \ \middle| \ \lambda_{\max}(\nabla^2 \mathcal{L}(\theta)) \leq \frac{2e^\epsilon}{\eta} \right\}. \tag{4}$$

To simplify the presentation, we focus on the case with $\epsilon = 0$ w.l.o.g.[1] and unless otherwise specified, a *"stable solution"* refers to an element of $\mathcal{F}(\eta, 0, \mathcal{D})$ in the remainder of the paper.

For the data generation process, we will consider two settings of interest: (1) the fixed design nonparametric regression setting (with noisy labels) (2) the agnostic statistical learning setting. They have different data assumptions and performance metrics to capture "generalization".

**Nonparametric Regression with Noisy labels.** In this setting, we assume fixed input $x_1, \cdots, x_n$ and $y_i = f_0(x_i) + \epsilon_i$ for $i \in [n]$, where $f_0 : \mathbb{R} \to \mathbb{R}$ is the ground-truth (target) function and $\{\epsilon_i\}_{i=1}^n$ are independent Gaussian noises $\mathcal{N}(0, \sigma^2)$. Our goal is find a ReLU NN $f$ using the dataset to minimize the mean squared error (MSE):

$$\text{MSE}(f) = \frac{1}{n} \sum_{i=1}^n \left( f(x_i) - f_0(x_i) \right)^2. \tag{5}$$

It is nonparametric because we do not require $f_0$ to be described by a smaller number of parameters, but rather satisfy certain regularity conditions. Specifically, we focus on estimating target functions inside the first order bounded variation class

$$f_0 \in \text{BV}^{(1)}(B, C_n) := \left\{ f : [-x_{\max}, x_{\max}] \to \mathbb{R} \ \Big| \ \max_x |f(x)| \le B, \int_{-x_{\max}}^{x_{\max}} |f''(x)| dx \le C_n \right\},$$

where $f''$ denotes the second-order weak derivative of $f$ and we define a short hand $\text{TV}^{(1)}(f) := \int_{-x_{\max}}^{x_{\max}} |f''(x)| dx$, which we refer to as the $\text{TV}^{(1)}$ (semi)norm of $f$ throughout the paper. We refer readers to a recent paper [Hu et al., 2022, Section 1.2] for the historical importance and the challenges in estimating the BV functions. The complexity of such function class is discussed in Appendix D.

**Agnostic statistical learning and generalization gap.** In this setting, we assume the $n$ data points $\{(x_i, y_i)\}_{i=1}^n$ are drawn i.i.d. from an unknown distribution $\mathcal{P}$ defined on $[-x_{\max}, x_{\max}] \times [-D, D]$. The expected performance on new data points is called "Risk", $R(f) = \mathbb{E}_{(x,y) \sim \mathcal{P}}[\ell(f, (x, y))]$. We define the absolute difference between training loss and the risk:

$$\text{Gen}(f) := \text{GeneralizationGap}(f) = |R(f) - \mathcal{L}(f)|.$$

We say that $f$ generalizes if $\text{GeneralizationGap}(f) \to 0$ as $n \to \infty$ with high probability.

## 3 Stable Solutions Cannot Interpolate Noisy Labels

A large portion of previous works studying minima stability assume the learned function interpolates the data. However, for various optimization problems, it is unclear whether there exists such an interpolating solution that is stable, especially when the number of samples $n$ becomes large.

For the nonparametric regression problem with noisy labels, we design an example where any interpolating function can not be stable. Before presenting the example, we first define the $g$ function, which will be the weight function of the weighted $\text{TV}^{(1)}$ norm throughout the paper: for $x \in [-x_{\max}, x_{\max}]$, $g(x) = \min\{g^-(x), g^+(x)\}$ with

$$\begin{aligned} g^-(x) &= \mathbb{P}^2(X < x)\mathbb{E}[x - X | X < x]\sqrt{1 + (\mathbb{E}[X | X < x])^2}, \\ g^+(x) &= \mathbb{P}^2(X > x)\mathbb{E}[X - x | X > x]\sqrt{1 + (\mathbb{E}[X | X > x])^2}, \end{aligned} \tag{6}$$

where $X$ is drawn from the empirical distribution of the data (a sample chosen uniformly from $\{x_j\}$).

For various distributions of training data (*e.g.* Gaussian distribution, uniform distribution), most of $g$'s mass is located at the center while $g$ decays towards the extreme data points. The same $g(x)$ is also applied as the weight function in Mulayoff et al. [2021], where they derived an upper bound $\int_{-x_{\max}}^{x_{\max}} |f''(x)| g(x) dx \le \frac{1}{\eta} - \frac{1}{2}$ assuming $f$ is linearly stable (Definition 2.1) and interpolating. We generalize the same upper bound to all stable solutions in $\mathcal{F}(\eta, 0, \mathcal{D})$ as in Theorem C.2. Below we construct a counter-example, where we can prove a contradicting lower bound of $\int_{-x_{\max}}^{x_{\max}} |f''(x)| g(x) dx$ for any interpolating $f$, thus disproving the assumption of interpolation.

**Counter-example.** We fix $x_i = \frac{2x_{\max} i}{n-1} - \frac{(n+1)x_{\max}}{n-1}$ for $i \in [n]$ and $f_0(x) = 0$ for any $x$, which implies that $y_i$'s are independent random variables from $\mathcal{N}(0, \sigma^2)$.

---

[1]To handle the case when $\epsilon > 0$, just replace $\eta$ with $\eta e^{-\epsilon}$ in all bounds in the remainder of the paper.

**Theorem 3.1.** *For the counter-example, with probability $1 - \delta$, for any interpolating function $f$,*

$$\int_{-x_{\max}}^{x_{\max}} |f''(x)|g(x)dx = \Omega\left(\sigma n\left[n - 24\log\left(\frac{1}{\delta}\right)\right]\right), \tag{7}$$

*where the randomness comes from the noises $\{\epsilon_i\}$. Under this high-probability event, when $n \geq \Omega\left(\sqrt{\frac{1}{\sigma\eta}}\log\left(\frac{1}{\delta}\right)\right)$, any stable solution $f$ for GD with step size $\eta$ will not interpolate the data, i.e.*

$$\mathcal{F}(\eta, 0, \mathcal{D}) \cap \{f \mid f(x_i) = y_i, \; \forall\, i \in [n]\} = \emptyset. \tag{8}$$

The proof of Theorem 3.1 is deferred to Appendix F due to space limit. This result, together with Mulayoff et al. [2021, Theorem 1], implies that gradient descent cannot converge to interpolating solutions unless $\eta = O(1/n^2)$. It also implies (when combined with Theorem 4.1) an intriguing geometric insight that all twice-differentiable interpolating solutions must be very sharp, *i.e.*, its largest eigenvalue is larger than $\Omega(n^2)$ (see details in Appendix J ). Moreover, we highlight that the conclusion of Theorem 3.1 is consistent with our observation in Figure 2(a), where the learned function tends to be smoother and would not interpolate the data as $\eta$ becomes larger. Therefore, in the following discussion, we consider the case without assuming interpolation.

# 4 Main Results

In this section, we present the main results about stable solutions for GD (functions in $\mathcal{F}(\eta, 0, \mathcal{D})$) from three aspects. Section 4.1 describes the implicit bias of stable solutions of gradient descent with large learning rate in the function space. Section 4.2 and 4.3 derive concrete generalization bounds that leverage the implicit biases in the *distribution-free statistical learning* setting and the *non-parametric regression* setting respectively. An outline of the proof of our main theorems is given in Appendix B. The full proof details are deferred to the appendix.

## 4.1 Implicit Bias of Stable Solutions in the Function Space

We begin with characterizing the stable solutions for GD with step size $\eta$ without the assumption of interpolation (there can be $i \in [n]$ such that $f(x_i) \neq y_i$). Similar to the interpolating case, the learned stable function $f$ enjoys a (weighted) $\mathrm{TV}^{(1)}$ bound as below.

**Theorem 4.1.** *For a function $f = f_\theta$ where the training (square) loss $\mathcal{L}$ is twice differentiable at $\theta$,[2]*

$$\int_{-x_{\max}}^{x_{\max}} |f''(x)|g(x)dx \leq \frac{\lambda_{\max}(\nabla_\theta^2 \mathcal{L}(\theta))}{2} - \frac{1}{2} + x_{\max}\sqrt{2\mathcal{L}(\theta)}, \tag{9}$$

*where $g(x)$ is defined as (6). Moreover, if we assume $y_i = f_0(x_i) + \epsilon_i$ for independent noise $\epsilon_i \sim \mathcal{N}(0, \sigma^2)$, then with probability $1 - \delta$ where the randomness is over the noises $\{\epsilon_i\}$,*

$$\int_{-x_{\max}}^{x_{\max}} |f''(x)|g(x)dx \leq \frac{\lambda_{\max}(\nabla_\theta^2 \mathcal{L}(\theta))}{2} - \frac{1}{2} + \widetilde{O}\left(\sigma x_{\max} \cdot \min\left\{1, \sqrt{\frac{k}{n}}\right\}\right) + x_{\max}\sqrt{\mathrm{MSE}(f)}. \tag{10}$$

*In addition, if $f = f_\theta$ is a stable solution of GD with step size $\eta$ on dataset $\mathcal{D}$, i.e., $f_\theta \in \mathcal{F}(\eta, 0, \mathcal{D})$ as in (4), then we can replace $\frac{\lambda_{\max}(\nabla_\theta^2 \mathcal{L}(\theta))}{2}$ with $\frac{1}{\eta}$ in (9) and (10).*

Theorem 4.1, which we prove in Appendix G, associates the local curvature of the loss landscape at $\theta$ with the smoothness of the function $f_\theta$ it represents as measured in a weighted $\mathrm{TV}^{(1)}$ norm. In short, it says that *flat solutions are simple*. The result is a strict generalization of Theorem 1 in Mulayoff et al. [2021] which requires interpolation, i.e., $\mathcal{L}(\theta) = 0$. Observe that the number of neurons $k$ does not appear in (9) and have no effect in (10) when $k > n$, thus the result applies to arbitrarily overparameterized two-layer NNs. Under the standard nonparametric regression assumption, (10) is a stronger bound that asymptotically matches the bound under interpolation [Mulayoff et al., 2021, Theorem 1] when $k = o(n)$ and $\mathrm{MSE}(f_\theta) = o(1)$ as the number of data points $n \to \infty$.

Another interesting observation when combining (10) with Theorem 3.1 is that for all interpolating solutions (observe that $\mathrm{MSE}(f_\theta) = \widetilde{O}(\sigma^2)$ w.h.p.)

$$\lambda_{\max}(\nabla_\theta^2 \mathcal{L}(\theta)) \geq \Omega(n^2\sigma) - \widetilde{O}(\sigma).$$

---

[2]W.l.o.g. we assume that $x_{\max} \geq 1$. If this does not hold, we can directly replace $x_{\max}$ in the bounds by 1.

To say it differently, *all interpolating solutions are very sharp minima when the labels are noisy*. This provides theoretical explanation of the empirical observation that noisy labels are harder to overfit using gradient training [Zhang et al., 2021].

Note that we leave the term $\mathcal{L}(\theta)$ (or $\mathrm{MSE}(f)$) in the $\mathrm{TV}^{(1)}$ bound. Therefore, we can plug any upper bound for these terms into Theorem 4.1 for a concrete result, and below we instantiate the $\mathrm{TV}^{(1)}$ bound with a crude MSE bound under the assumption that $f$ is "optimized".

**Corollary 4.2.** *In the nonparametric regression problem with ground-truth function $f_0$, for a stable solution $f = f_\theta$ of GD with step size $\eta$ where $\mathcal{L}$ is twice differentiable at $\theta$, assume that $f$ is optimized, i.e, the empirical loss of $f$ is smaller than $f_0$: $\frac{1}{2n} \sum_{i=1}^{n} (f(x_i) - y_i)^2 \leq \frac{1}{2n} \sum_{i=1}^{n} (f_0(x_i) - y_i)^2$, then with probability $1 - \delta$, the function $f$ satisfies*

$$\int_{-x_{\max}}^{x_{\max}} |f''(x)| g(x) dx \leq \frac{1}{\eta} - \frac{1}{2} + \widetilde{O}\left(\sigma x_{\max}\right), \tag{11}$$

*where the randomness is over the noises $\{\epsilon_i\}$ and $\widetilde{O}$ suppresses logarithmic terms of $n, 1/\delta$.*

The assumption of optimized $f$ is rather mild since, in practice, gradient-based optimizers are commonly quite effective in loss minimization (see also our experiments). With such assumption, we can derive an MSE upper bound (with high probability) of order $\widetilde{O}(\sigma^2)$ (details in Lemma G.5), and thus the $\mathrm{TV}^{(1)}$ bound above.

In some cases, we can decrease the MSE upper bound we assumed in Corollary 4.2, and use this to improve the resulting $\mathrm{TV}^{(1)}$ bound (11). For instance, if the neural network is under-parameterized (i.e. $k$ is smaller than $n$), in Appendix G.2 we derive a (high-probability) bound for MSE of order $\widetilde{O}(k/n)$, which implies that the last term in (11) becomes $\widetilde{O}(\sigma x_{\max} \sqrt{k/n})$. The term vanishes if $n/k$ is large enough, where the $\mathrm{TV}^{(1)}$ bound reduces to the noiseless and interpolating case.

## 4.2 GD on ReLU NN Does Not Overfit

Why would anyone care about the function space implications of stable solutions? The next theorem shows that these solutions cannot overfit (in the strict interior of the data support) without making strong assumptions on the shape of the data distribution.

**Theorem 4.3.** *Let $\mathcal{P}$ be a joint distribution of $(x, y)$ supported on $[-x_{\max}, x_{\max}] \times [-D, D]$. Assume the dataset $\mathcal{D} \sim \mathcal{P}^n$ i.i.d. For any fixed interval $\mathcal{I} \subset [-x_{\max}, x_{\max}]$ and a universal constant $c > 0$ such that with probability $1 - \delta/2$, $g(x) \geq c$ for all $x \in \mathcal{I}$, if the function $f = f_\theta$ is a stable solution of GD with step size $\eta$ such that $\mathcal{L}$ is twice differentiable at $\theta$ and $\|f\|_\infty \leq D$, with probability $1 - \delta$ (randomness over the dataset), the generalization gap restricted to $\mathcal{I}$ satisfies*

$$\mathrm{Gen}_{\mathcal{I}}(f) := \left| \mathbb{E}_{\mathcal{I}} \left[ (f(x) - y)^2 \right] - \frac{1}{n_{\mathcal{I}}} \sum_{x_i \in \mathcal{I}} (f(x_i) - y_i)^2 \right| \leq \widetilde{O}\left( D^{\frac{9}{5}} \left[ \frac{x_{\max} \left( \frac{1}{\eta} - \frac{1}{2} + 2x_{\max}D \right)}{n_{\mathcal{I}}^2} \right]^{\frac{1}{5}} \right), \tag{12}$$

*where $\mathbb{E}_{\mathcal{I}}$ means that $(x, y)$ is a new sample from the data distribution conditioned on $x \in \mathcal{I}$ and $n_{\mathcal{I}}$ is the number of data points in $\mathcal{D}$ such that $x_i \in \mathcal{I}$.*

The proof of Theorem 4.3 is deferred to Appendix H. Briefly speaking, we show that in the strict interior of the data support, the generalization gap will vanish as the number of data points $n$ increases. This vanishing generalization gap further implies that the expected performance on new data points is close to the (observable) training loss, i.e., the output stable solutions do not overfit.

Regarding our assumptions, in addition to standard boundedness assumptions, we focus on the strict interior of the domain where $g$ can be lower bounded (i.e. interval $\mathcal{I}$). This is because for extreme data points, $g(x)$ decays and thus imposes little constraint on the output function $f$. In Appendix H.1, we show that if the marginal distribution of $x$ is the uniform distribution on $[-x_{\max}, x_{\max}]$, when $n$ is sufficiently large, $\mathcal{I}$ can be chosen as $[-\frac{2x_{\max}}{3}, \frac{2x_{\max}}{3}]$. In this case, with high probability, $\mathcal{I}$ incorporates a large portion of the data points and $n_{\mathcal{I}} = \Omega(n)$. More illustrations about the choice of $\mathcal{I}$ under various data distributions are deferred to Appendix H.2.

Meanwhile, the generalization gap bound has dependence $\frac{1}{\eta}$ on the learning rate $\eta$. Therefore, as we increase the learning rate (in a reasonable range), the learned stable solution tends to be smoother, which further implies better generalization performances.

### 4.3 GD on ReLU NN Achieves Optimal Rate for Estimating BV(1) Functions

Finally, we zoom into the nonparametric regression task that we described in Section 2 where $x_1, \cdots, x_n$ are fixed and the noisy labels $y_i = f_0(x_i) + \mathcal{N}(0, \sigma^2)$ independently for $i \in [n]$ for a ground truth function $f_0$ in the first order bounded variation class (see Section 2 for details). Similar to Theorem 4.3, we focus on the strict interior $\mathcal{I}$ of $[-x_{\max}, x_{\max}]$, but instead of a generalization gap bound, we prove an MSE bound against $f_0$ on $\mathcal{I}$ that nearly matches the theoretical limit.

**Theorem 4.4.** *Under the same conditions in Corollary 4.2, for any interval $\mathcal{I} \subset [-x_{\max}, x_{\max}]$ and a universal constant $c > 0$ such that $g(x) \geq c$ for all $x \in \mathcal{I}$ and $f$ is optimized over $\mathcal{I}$, i.e. $\sum_{x_i \in \mathcal{I}} (f(x_i) - y_i)^2 \leq \sum_{x_i \in \mathcal{I}} (f_0(x_i) - y_i)^2$, if the output stable solution $\theta$ satisfies $\|\theta\|_\infty \leq \rho$ (for some constant $\rho > 0$) and the ground truth $f_0 \in \mathrm{BV}^{(1)}(k\rho^2, \frac{1}{c}\widetilde{O}(\frac{1}{\eta} + \sigma x_{\max}))$, then with probability $1 - \delta$ (over the random noises $\{\epsilon_i\}$), the function $f = f_\theta$ satisfies*

$$\mathrm{MSE}_{\mathcal{I}}(f) = \frac{1}{n_{\mathcal{I}}} \sum_{x_i \in \mathcal{I}} (f(x_i) - f_0(x_i))^2 \leq \widetilde{O}\left(\left(\frac{\sigma^2}{n_{\mathcal{I}}}\right)^{\frac{4}{5}} \left(\frac{x_{\max}}{\eta} + \sigma x_{\max}^2\right)^{\frac{2}{5}}\right), \quad (13)$$

*where $n_{\mathcal{I}}$ is the number of data points in $\mathcal{D}$ such that $x_i \in \mathcal{I}$.*

The proof of Theorem 4.4 is deferred to Appendix I. Below we discuss some interesting aspects of the result. First of all, Theorem 4.4 focuses on an interval $\mathcal{I}$ where $g(x)$ can be lower bounded. In this way, we ignore the extreme data points and derive an MSE upper bound (restricted to $\mathcal{I}$) of order $\widetilde{O}(n_{\mathcal{I}}^{-4/5})$, which matches the minimax optimal rate for estimating $\mathrm{BV}^{(1)}$ functions [see, e.g., Donoho and Johnstone, 1998, Theorem 1]. In contrast, it is well-known that neural networks in the kernel regime (and any other "linear smoothers") must incur a strictly suboptimal worst-case $\mathrm{MSE} = \Omega(n_{\mathcal{I}}^{-2/3})$ [Donoho et al., 1990, Suzuki, 2018]. According to the discussions in Appendix H.1, if the data follows a uniform distribution, the interval $\mathcal{I}$ can incorporate most of the data points where $n_{\mathcal{I}} = \Omega(n)$.

Meanwhile, the MSE bound has dependence $\eta^{-2/5}$ on the learning rate $\eta$. Such dependence is because with a larger learning rate, the learned function $f$ will have smaller $\mathrm{TV}^{(1)}$ bound, and therefore the set of possible output functions will contain fewer non-smooth functions, which implies a tighter MSE bound. However, this does not mean that a larger learning rate is always better. When $\eta$ is too large, GD may diverge, and even if it does not, the set of stable solutions cannot approximate the ground truth $f_0$ well if $\int_{-x_{\max}}^{x_{\max}} |f_0''(x)| dx \gg \frac{1}{c\eta} + \frac{\sigma x_{\max}}{c}$, thus failing to satisfy the "optimized" assumption. In our experiments, we verify the "optimized" assumption numerically for a wide range of $\eta$ and demonstrate that by tuning learning rate $\eta$, we are adapting to the unknown $\mathrm{TV}^{(1)}(f_0)$.

Lastly, we remark that Theorem 4.4 holds for arbitrary $k$, even if the neural network is heavily over-parameterized ($k \gg n$). The dependence $\frac{1}{\eta} + \sigma x_{\max}$ on $\eta$ results from the $\mathrm{TV}^{(1)}$ bound in Corollary 4.2, where the term $\frac{1}{\eta}$ will dominate if $\eta \leq \frac{1}{\sigma x_{\max}}$, which is the case if the step size is not large. Furthermore, if $\frac{n}{k}$ is sufficiently large, we can improve the $\mathrm{TV}^{(1)}$ bound in Corollary 4.2 as in Appendix G.2, and improve the mean squared error $\mathrm{MSE}_{\mathcal{I}}(f)$ to $\widetilde{O}\left(\left(\frac{\sigma^2}{n_{\mathcal{I}}}\right)^{\frac{4}{5}} \left(\frac{x_{\max}}{\eta}\right)^{\frac{2}{5}}\right)$ accordingly. The assumptions and detailed statements for the improved MSE are deferred to Appendix I.1.

## 5 Experiments

In this section, we empirically validate our claims by training a two-layer fully connected neural network with ReLU activation using gradient descent (GD) with varying step sizes. We focus on fitting a mildly overparameterized ReLU network to a simple nonparametric regression problem. The input dataset comprises of 30 equally spaced fixed design points $\{x_i\}_{i=1}^n$, where each $x_i \in [-0.5, 0.5]$ ($n = 30$). Label $y_i = f_0(x_i) + \mathcal{N}(0, \sigma^2)$ with $\sigma = 0.5$ and $f_0(x) = (2x+1)\mathbf{1}(x \leq 0) + (-2x+1)\mathbf{1}(x > 0)$.

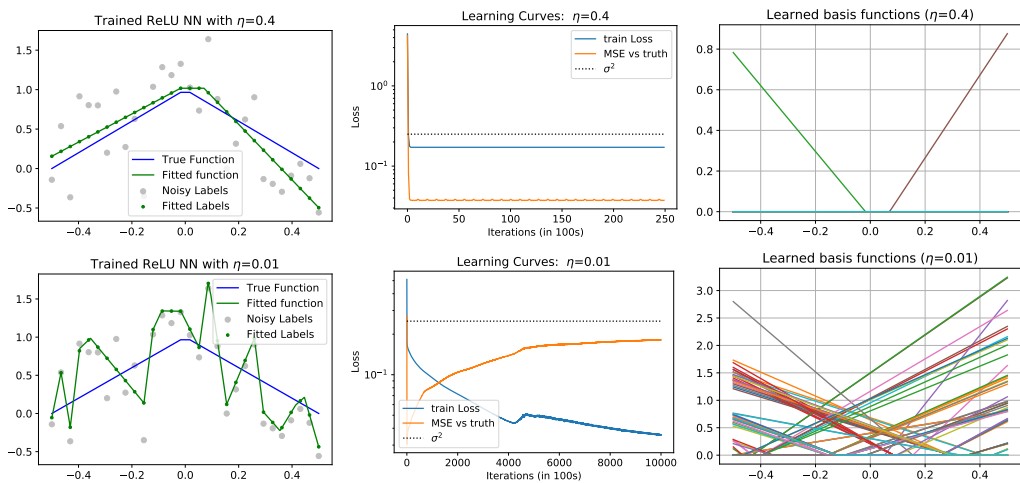

Figure 3: Highlights of our numerical simulation for large step size ($\eta = 0.4$, **first row**) and small step size ($\eta = 0.01$, **second row**) gradient descent training of a univariate ReLU NN with $n = 30$ noisy observations and $k = 100$ hidden neurons. From left to right, the three columns illustrate (a) Trained NN function (b) Learning curves (c) Learned basis functions (each of the 100 neurons).

The two-layer ReLU network is parameterized by $\theta$ (see Section 2) with $k = 100$ neurons per layer. The network uses standard parameterization (scale factor of 1) and parameters are initialized randomly (see Figure 7 for the initial basis functions).

Figure 2 (in the introduction) illustrates how changing the learning rate affects the learned ReLU NN that GD-training stabilizes on. The main take-aways are (a) large learning rate learns flatter minima which represent more regular functions (in $\text{TV}^{(1)}$); (b) Our bound from Theorem 4.1 is a very accurate description of the curvature of the Hessian as well as a valid upper bound of the $\text{TV}^{(1)}$-(pseudo) norm; (c) When we tune learning rate $\eta$, it is implicitly regularizing the complexity, which provides a satisfying variance tradeoff explanation to how GD-training works.

Figure 3 provides further details on the learning curves and representation learning. We note that the learned representation is very different from the initialization, thus our experiments are clearly describing phenomena not covered by the "kernel" regime. In addition, it seems that all solutions that GD finds after a small number of iterations satisfy the "optimized" assumption as required in Theorem 4.4. In the appendix (Figure 8), we provide empirical justification for the other assumption we make about the twice-differentiability of the solutions. More experiments can be found in the appendix with more learning rate choices, as well as a discussion on the *catastrophic* and *tempered* overfitting of interpolating solutions when we adjust $k$.

## 6 Conclusion

In this paper, we took a new look into how gradient descent-trained two-layer ReLU neural networks generalize from a lens of minima stability (and the closely related Edge-of-Stability phenomena). We focused on univariate inputs with noisy labels and showed GD with typical choice of learning rate cannot interpolate the data. We also established that local smoothness of the training loss functions implies a first order total variation constraint on the function the neural network represents, hence proving that all such solutions have a vanishing generalization gap inside the strict interior of the data support. In addition, under a mild assumption, we prove that these stable solutions achieve near-optimal rate for estimating first-order bounded variation functions. Future work includes generalization beyond 1D input, two-hidden layers, and understanding the choice of optimization algorithms.

## Acknowledgments

The research is partially supported by NSF #2134214. DQ, KZ, ES and YW's work was partially completed while they were with the Department of Computer Science at UCSB. The research of DS was Funded by the European Union (ERC, A-B-C-Deep, 101039436). Views and opinions expressed are however those of the author only and do not necessarily reflect those of the European Union or the European Research Council Executive Agency (ERCEA). Neither the European Union nor the granting authority can be held responsible for them. DS also acknowledges the support of the Schmidt Career Advancement Chair in AI.

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

# A   Full Experimental Results

## A.1   Stable Minima GD Converges to and Learning Curves

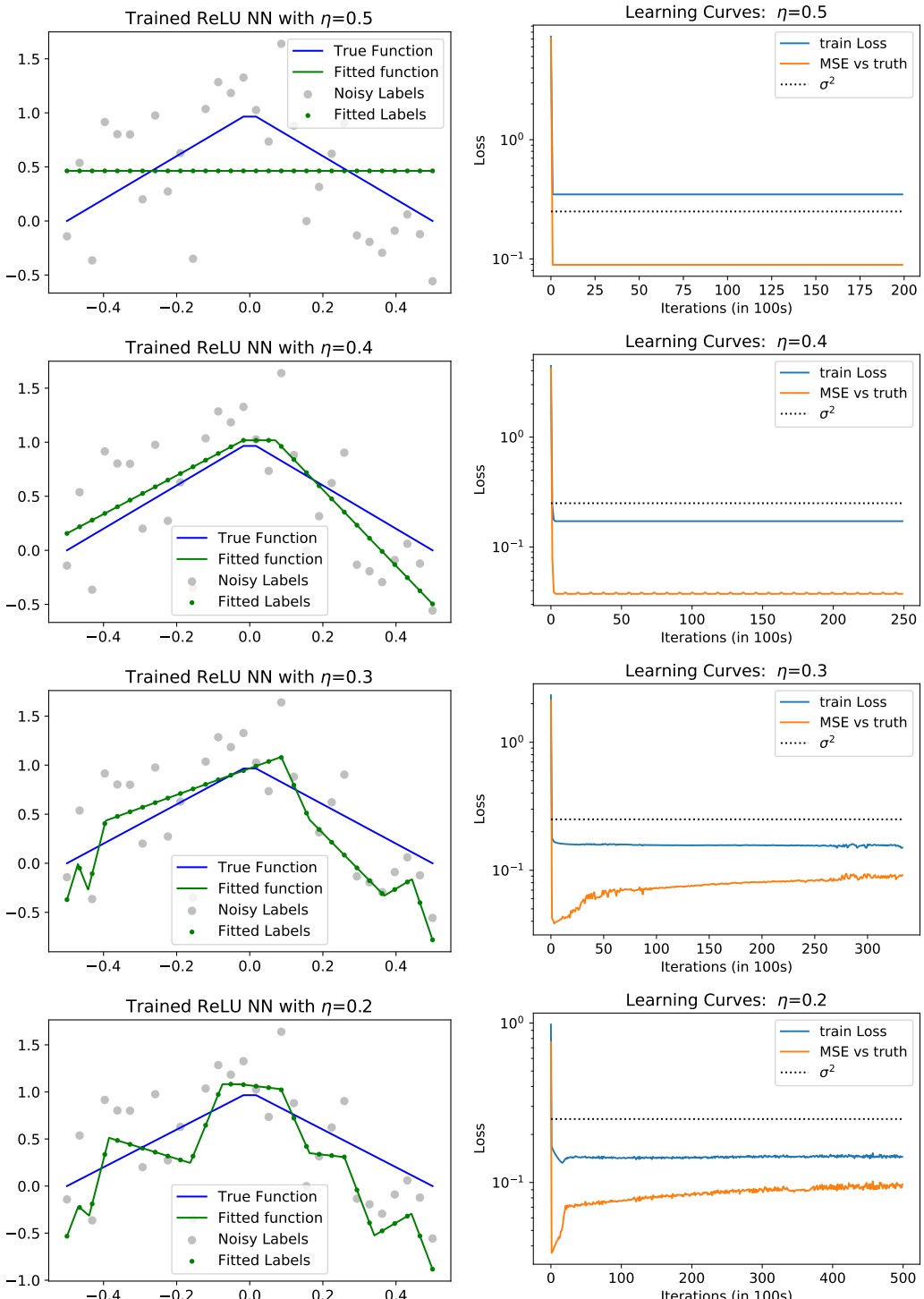

Figure 4: Illustration of the solutions gradient descent with learning rate $\eta$ converges to (Part I). As $\eta$ decreases, the fitted function goes from simple to complex. Any line below the $\sigma^2$ line **satisfies the "optimized" assumption** from Corollary 4.2 and Theorem 4.4.

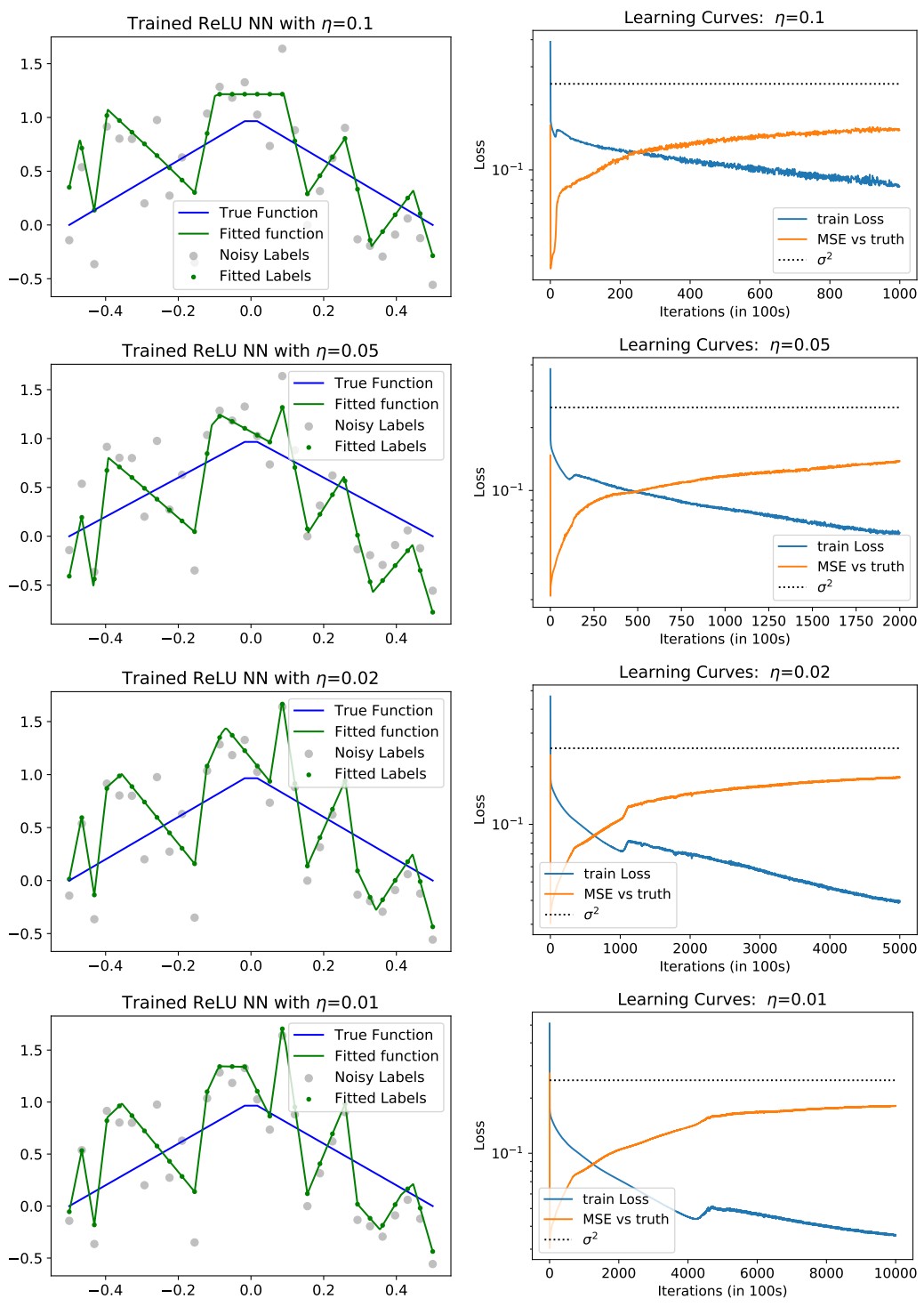

Figure 5: Illustration of the solutions gradient descent with learning rate $\eta$ converges to (Part II). As $\eta$ decreases further, the fitted function starts to overfit to the noisy label.

## A.2 Interpolating Solutions as the Number of Hidden Neurons Increases

In this section, we illustrate a sequence of interpolating solutions that are the global optimal solutions, which is also the kernel limit in the "lazy" regime. The results are obtained by randomly initializing

the weights, but solve the minimum norm solution by directly solving the least square problem (optimizing only the second layer weights.)

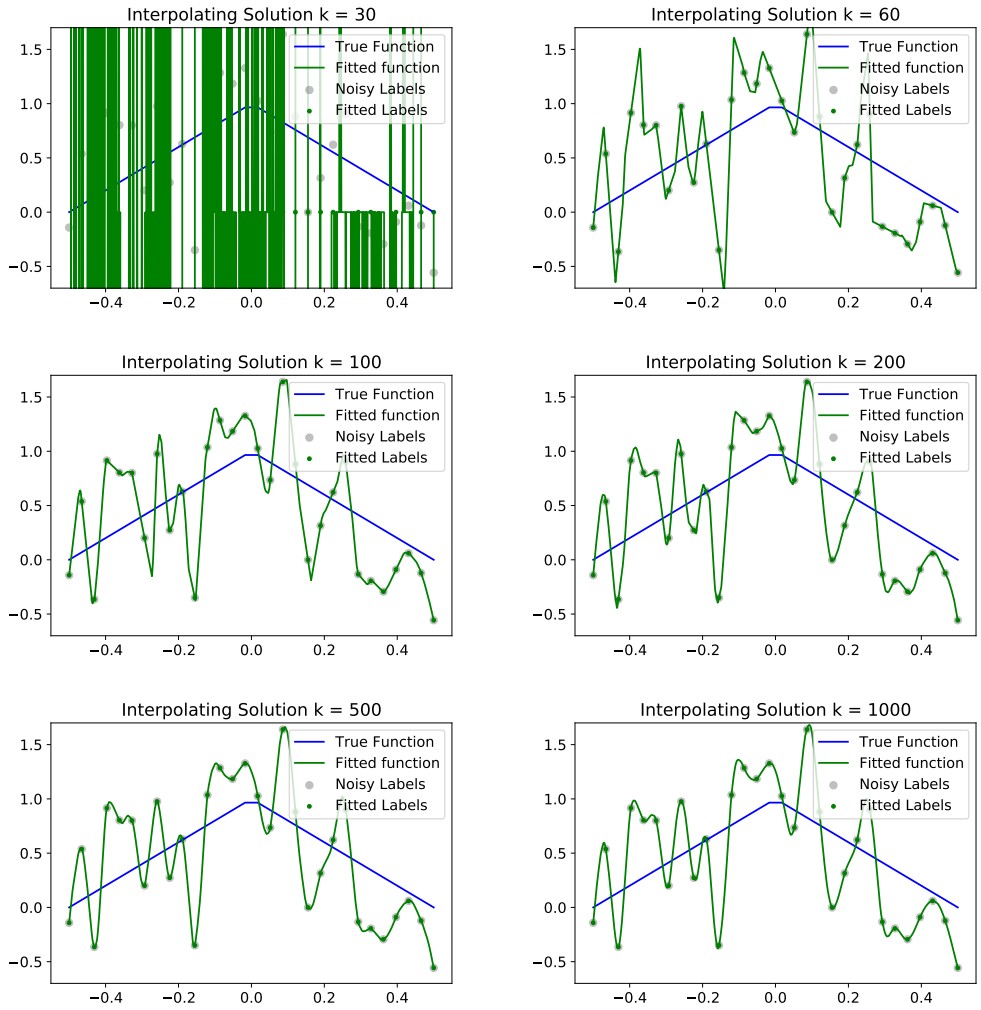

Figure 6: Examples of global optimal (interpolating) solutions (fitting only second layer weights). Notice that the number of data points $n = 30$. When the model is barely able to interpolate ($k = 30$), the fitted function experiences the *catastrophic* overfitting. When the number of neurons $k$ increases the interpolation solution becomes smoother and enters the *tempered* overfitting regime.

## A.3 Representation Learning of Large Learning Rate: Visualizing Learned Basis Functions.

In this section, we visualize the basis functions at initialization and after training with different learning rate.

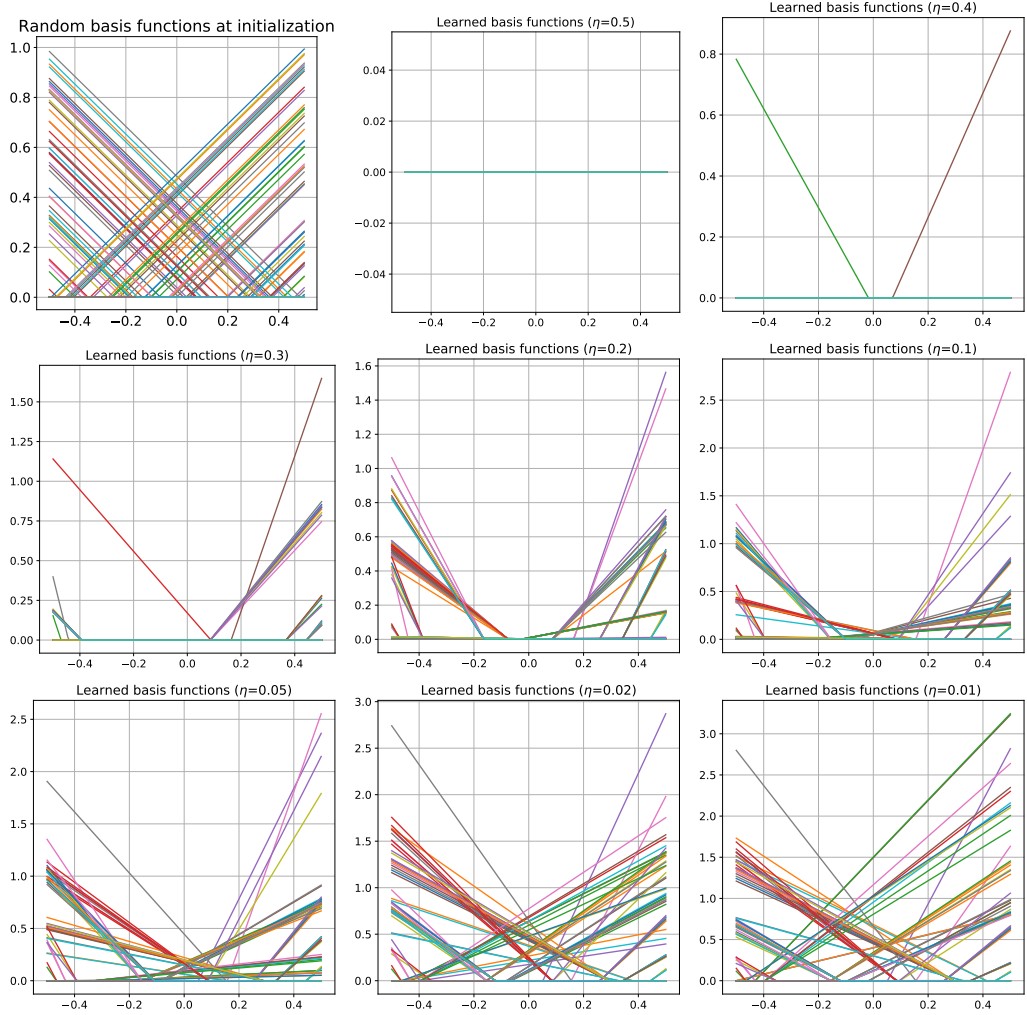

Figure 7: Illustration of the learned basis function with learning rate $\eta$. It is clear from the figures that there were substantial representation learning and the number of active basis functions gets smaller as the learning rate $\eta$ gets bigger.

We make several observations about Figure 7. First, the learned basis functions are very different from the initialization, so a lot of representation learning is happening, in comparison to the "kernel" regime in which nearly no representation learning is happening. Second, as $\eta$ gets smaller, the number of learned basis functions increases, hence increasing the number of knots in the fitted function. Third, the learned basis function displays a strong "clustering" effect in the sense that despite overparameterization, many learned basis functions end up being the same on the data support. Interestingly, they are not the same on $\mathbb{R}$, we verified that they are still different outside the data support, e.g., one of the learned basis function has a knot at $x < -800$.

## A.4 Knots of the Learned ReLU NN (aka Linear Splines) and Their Coefficients

Recall that a linear spline is a continuous piecewise linear function and a two-layer ReLU NN with $k$ neurons span the class of all linear splines with at most $k$ knots. In Figure 8, we visualize the locations of the knots of the linear spline that the learned ReLU neural networks represent.

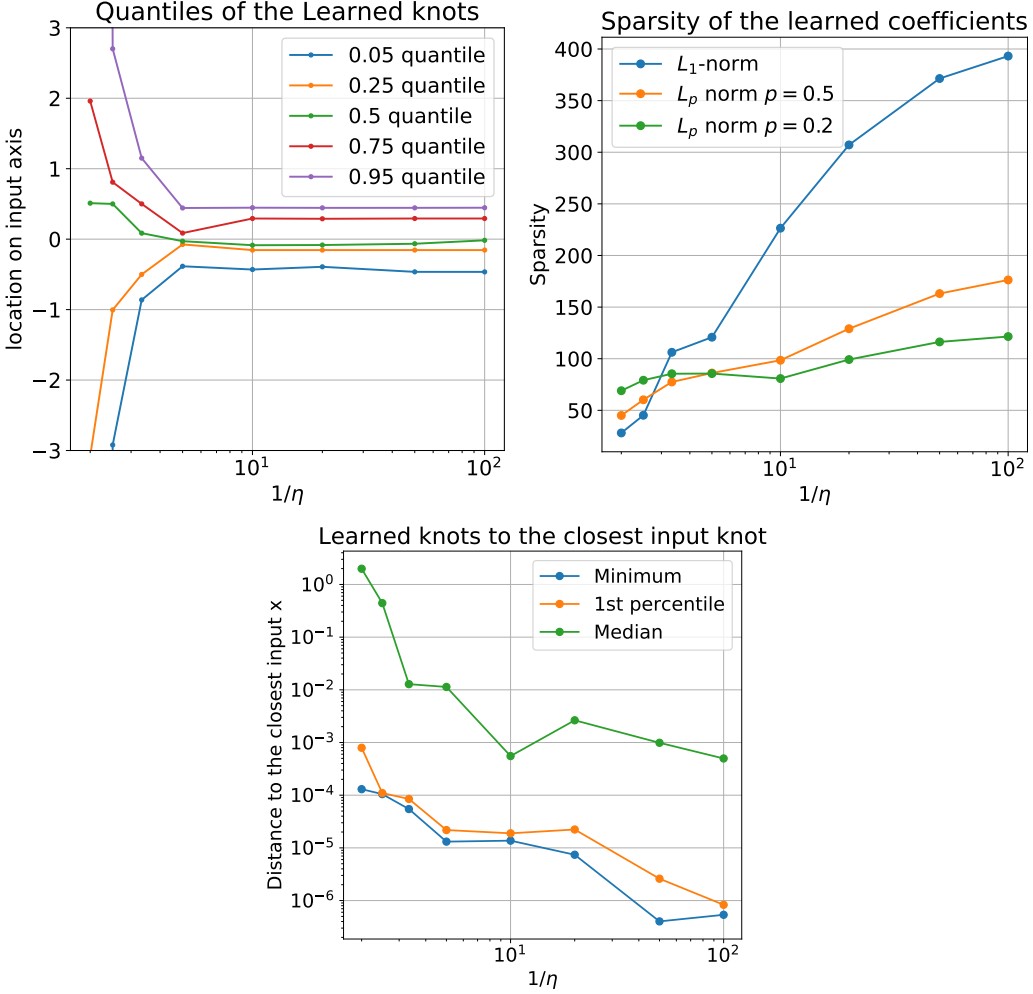

Figure 8: Illustration of the learned (steady-state) ReLU NN with different learning rate. Recall that all ReLU NNs are linear splines, therefore the location of the knots (i.e., the change points of the linear pieces) describes the representation learning that happens. Each basis function is a ReLU function at the knot. The final ReLU NN is a linear combination of these learned basis functions. In the first panel, we plot the quantiles of the locations of the learned knots as the function of $1/\eta$. In the second panel, we plot the sparsity of the learned coefficients in sparse $L_1$ and $L_p$ norm as the function of $1/\eta$. The third panel plots the distance of the learned knots from the closest input data points. **This empirically verifies that the solution that gradient descent finds at the end is a twice differentiable function** w.r.t. the parameters in the sense that not a single learned knot is exactly at the input data point, thus ensuring the applicability of our theorems.

In the first panel of Figure 8, we find for large learning rate, most of the location of the learned knot is actually outside the data support on $[-0.5, 0.5]$. This is a somewhat surprising finding in that the mechanism of neural network learning may actually be "pushing the knot outside the data support" so they become inactive on the training data (and only the ReLU truncated 0s are active). This is a new (and very interesting) way to understand how sparsity arrives in gradient descent learning.

The second panel describes the sparsification effect of the implicit biases from large learning rate, which again, indicates that the weighted TV1 constraint is indeed making the learned function sparse (in the coefficient vector). The third panel shows that despite that the learning rate gets as small as $1e-2$, none of the learned basis function actually have knots coinciding with any of the input data, thus empirically justifying our assumptions on the twice-differentiability of the solutions GD finds.

# B  Proof Overview

In this section, we outline proof ideas for the main theorems in Section 4.

**Proof overview of Theorem 4.1.** According to direct calculation, the Hessian is given by

$$\nabla_\theta^2 \mathcal{L}(\theta) = \frac{1}{n}\sum_{i=1}^n (\nabla_\theta f_\theta(x_i))(\nabla_\theta f_\theta(x_i))^T + \frac{1}{n}\sum_{i=1}^n (f_\theta(x_i) - y_i)\nabla_\theta^2 f(x_i). \tag{14}$$

Let $v$ denote the unit eigenvector ($\|v\|_2 = 1$) of $\frac{1}{n}\sum_{i=1}^n(\nabla_\theta f_\theta(x_i))(\nabla_\theta f_\theta(x_i))^T$ with respect to the largest eigenvalue, then it holds that

$$\lambda_{\max}(\nabla_\theta^2\mathcal{L}(\theta)) \geq v^T\nabla_\theta^2\mathcal{L}(\theta)v$$

$$= \underbrace{\lambda_{\max}\left(\frac{1}{n}\sum_{i=1}^n(\nabla_\theta f_\theta(x_i))(\nabla_\theta f_\theta(x_i))^T\right)}_{(\star)} + \underbrace{\frac{1}{n}\sum_{i=1}^n(f_\theta(x_i) - y_i)v^T\nabla_\theta^2 f(x_i)v}_{(\#)}. \tag{15}$$

For the term $(\star)$, we connect the maximal eigenvalue at $\theta$ to the (weighted) $\mathrm{TV}^{(1)}$ norm of the corresponding $f = f_\theta$. Let $g(x)$ be defined as (6), Mulayoff et al. [2021, Lemma 4] shows that (details in Lemma G.1):

$$(\star) = \lambda_{\max}\left(\frac{1}{n}\sum_{i=1}^n(\nabla_\theta f_\theta(x_i))(\nabla_\theta f_\theta(x_i))^T\right) \geq 1 + 2\int_{-x_{\max}}^{x_{\max}}|f''(x)|g(x)dx. \tag{16}$$

For the term $(\#)$, we can bound it by the training loss $\mathcal{L}(\theta)$ using Cauchy-Schwarz inequality and a *somewhat surprising* uniform upper bound of $v^T\nabla_\theta^2 f(x_i)v$ in Lemma E.1:

$$|(\#)| \leq \sqrt{\frac{1}{n}\sum_{i=1}^n(f_\theta(x_i) - y_i)^2} \cdot \sqrt{\frac{1}{n}\sum_{i=1}^n(v^T\nabla_\theta^2 f(x_i)v)^2} \leq 2x_{\max}\sqrt{2\mathcal{L}(\theta)}. \tag{17}$$

The first conclusion (9) of Theorem 4.1 is derived by combining the inequalities above. For the second conclusion (10), note that the term $(\#)$ can be further decomposed as:

$$(\#) = \underbrace{\frac{1}{n}\sum_{i=1}^n(f_0(x_i) - y_i)v^T\nabla_\theta^2 f(x_i)v}_{(i)} + \underbrace{\frac{1}{n}\sum_{i=1}^n(f_\theta(x_i) - f_0(x_i))v^T\nabla_\theta^2 f(x_i)v}_{(ii)}. \tag{18}$$

Similar to $(\#)$, the term $|(ii)|$ can be bounded by $2x_{\max}\sqrt{\mathrm{MSE}(f_\theta)}$ using Cauchy-Schwarz inequality. Under the data-generating assumption $y_i - f_0(x_i) = \epsilon_i \sim \mathcal{N}(0, \sigma^2)$ i.i.d., $|(i)|$ can be bounded by a certain empirical *Gaussian complexity* term $\sup_\theta \frac{1}{n}\sum_i \epsilon_i h_\theta(x_i)$ with $h_\theta(x_i) = v^T\nabla_\theta^2 f(x_i)v$. The proof is complete by Lemma G.2 which bounds this Gaussian complexity term (w.h.p.) in two ways: (1) a *dimension-free* bound of $\widetilde{O}(\sigma x_{\max})$ and (2) $\widetilde{O}(\sigma x_{\max}\sqrt{k/n})$ for the under-parameterized case.

**Proof overview of Theorem 4.3.** Under the boundedness assumption in Theorem 4.3, we can prove a constant upper bound for $\int_{-x_{\max}}^{x_{\max}}|f''(x)|g(x)dx$ (Lemma H.1), which further implies another constant upper bound for $\int_\mathcal{I}|f''(x)|dx$. Therefore, the metric entropy (logarithmic of $\epsilon$-covering number $N_\epsilon$) of the possible output function class is of order $\epsilon^{-1/2}$ (Details in Lemma H.2).

For a fixed $\epsilon$-cover of the possible output function class, according to Hoeffding's inequality and a union bound, the uniform upper bound of $\mathrm{Gen}_\mathcal{I}(f)$ can be bounded by $\widetilde{O}(\sqrt{\log N_\epsilon/n_\mathcal{I}}) = \widetilde{O}(\epsilon^{-\frac{1}{4}}n_\mathcal{I}^{-\frac{1}{2}})$ with high probability. Note that the approximation error is of order $\widetilde{O}(\epsilon)$, therefore $\mathrm{Gen}_\mathcal{I}(f)$ can be uniformly bounded over the possible output function class by

$$\widetilde{O}(\epsilon) + \widetilde{O}(\epsilon^{-\frac{1}{4}}n_\mathcal{I}^{-\frac{1}{2}}) = \widetilde{O}(n_\mathcal{I}^{-\frac{2}{5}}), \tag{19}$$

where $\epsilon$ is chosen to minimize the bound. More details can be found in the proof of Theorem H.3.

**Proof overview of Theorem 4.4.** Similar to Theorem 4.3, we can prove a constant upper bound for $\int_\mathcal{I}|f''(x)|dx$, which implies that the metric entropy of the possible output function class (4) is of order $\epsilon^{-1/2}$ (Details in Lemma I.2). Therefore, the *critical radius $r$* is of order $\widetilde{O}(n_\mathcal{I}^{-\frac{2}{5}})$, which leads to a (high probability) MSE upper bound of order $\widetilde{O}(n_\mathcal{I}^{-\frac{4}{5}})$ using a self-bounding technique. More details about handling other parameters can be found in Appendix I.

# C  Some Optimization Results

**Lemma C.1** (Restate Lemma 2.2). *Consider the update rule in Definition 2.1, for any $\epsilon > 0$, a local minimum $\theta^\star$ is an $\epsilon$ linearly stable minimum of $\mathcal{L}$ if and only if $\lambda_{\max}(\nabla^2 \mathcal{L}(\theta^\star)) \leq \frac{2}{\eta}$.*

*Proof of Lemma C.1.* It holds that

$$
\begin{aligned}
\theta_{t+1} - \theta^\star &= \theta_t - \theta^\star - \eta \left( \nabla \mathcal{L}(\theta^\star) + \nabla^2 \mathcal{L}(\theta^\star)(\theta_t - \theta^\star) \right) \\
&= \theta_t - \theta^\star - \eta \nabla^2 \mathcal{L}(\theta^\star)(\theta_t - \theta^\star) \\
&= \left( I - \eta \nabla^2 \mathcal{L}(\theta^\star) \right) (\theta_t - \theta^\star),
\end{aligned}
\tag{20}
$$

where the first equation is from the update rule in Definition 2.1. The second equation holds because $\theta^\star$ is a local minimum and therefore $\nabla \mathcal{L}(\theta^\star) = 0$. As a result,

$$
\theta_t - \theta^\star = \left( I - \eta \nabla^2 \mathcal{L}(\theta^\star) \right)^t (\theta_0 - \theta^\star).
\tag{21}
$$

On one hand, if $\lambda_{\max}(\nabla^2 \mathcal{L}(\theta^\star)) \leq \frac{2}{\eta}$, it holds that

$$
\|\theta_t - \theta^\star\| \leq \left\| I - \eta \nabla^2 \mathcal{L}(\theta^\star) \right\|_2^t \cdot \|\theta_0 - \theta^\star\| \leq \|\theta_0 - \theta^\star\|,
\tag{22}
$$

where the second inequality is because all the eigenvalues of $I - \eta \nabla^2 \mathcal{L}(\theta^\star)$ is bounded between $[-1, 1]$. Therefore, $\theta^\star$ is $\epsilon$ linearly stable for any $\epsilon$.

On the other hand, if $\theta^\star$ is $\epsilon$ linearly stable, we choose $\theta_0$ such that $\frac{\theta_0 - \theta^\star}{\|\theta_0 - \theta^\star\|}$ is the top eigenvector of $\nabla^2 \mathcal{L}(\theta^\star)$ and $\|\theta_0 - \theta^\star\| = \epsilon$. Then we have

$$
\|\theta_t - \theta^\star\| = \left| 1 - \eta \lambda_{\max} \left( \nabla^2 \mathcal{L}(\theta^\star) \right) \right|^t \cdot \epsilon,
\tag{23}
$$

which implies that $\limsup_{t \to \infty} \left| 1 - \eta \lambda_{\max} \left( \nabla^2 \mathcal{L}(\theta^\star) \right) \right|^t \leq 1$, and therefore $\lambda_{\max}(\nabla^2 \mathcal{L}(\theta^\star)) \leq \frac{2}{\eta}$, which finishes the proof. $\square$

The following Theorem C.2 is an extension of the main result in Mulayoff et al. [2021]. Recall that stable solutions refer to the functions in $\mathcal{F}(\eta, 0, \mathcal{D})$ as defined in Section 2.

**Theorem C.2** (Extension of Theorem 1 in Mulayoff et al. [2021]). *Let $f = f_\theta$ be a stable solution for GD with step size $\eta$ where the training loss $\mathcal{L}(f) = 0$ and $\mathcal{L}$ is twice differentiable at $\theta$. Then*

$$
\int_{-x_{\max}}^{x_{\max}} |f''(x)| g(x) dx \leq \frac{1}{\eta} - \frac{1}{2},
\tag{24}
$$

*where $g(x) = \min\{g^-(x), g^+(x)\}$ for $x \in [-x_{\max}, x_{\max}]$ with*

$$
\begin{aligned}
g^-(x) &= \mathbb{P}^2(X < x) \mathbb{E}[x - X | X < x] \sqrt{1 + (\mathbb{E}[X|X < x])^2}, \\
g^+(x) &= \mathbb{P}^2(X > x) \mathbb{E}[X - x | X > x] \sqrt{1 + (\mathbb{E}[X|X > x])^2}.
\end{aligned}
\tag{25}
$$

*Here $X$ is drawn from the empirical distribution of the data (a sample chosen uniformly from $\{x_j\}$).*

*Proof of Theorem C.2.* The proof of Theorem 1 in Mulayoff et al. [2021] first proves that $\lambda_{\max}(\nabla^2 \mathcal{L}(\theta)) \leq \frac{2}{\eta}$ according to the assumption that $f$ is linearly stable, and then proves the conclusion above. Therefore, the same conclusion directly follows for the stable solutions $f = f_\theta$ in $\mathcal{F}(\eta, 0, \mathcal{D})$ satisfying $\lambda_{\max}(\nabla^2 \mathcal{L}(\theta)) \leq \frac{2}{\eta}$. $\square$

**Some discussions about the result.** Mulayoff et al. [2021] studied the problem assuming that the optimization converges to a global minimum and the learned function interpolates the training data, *i.e.* $\mathcal{L}(f) = 0$. In this way, they link the Hessian matrix to properties of the learned function $f$ and show that stable solutions of GD correspond to functions whose second order derivative has a bounded weighted norm. Moreover, as the learning rate increases, the set of stable solutions contains less and less non-smooth functions. For a fixed learning rate, according to the curve of $g(x)$, stable solutions tend to be smoother for instances near the center of the data distribution, and less smooth for instances near the edges. More discussions can be found in Mulayoff et al. [2021].

# D Bounded Variation Function Class and its Metric entropy

In this section, we first define the Besov function class. Then we recall the definition of bounded variation function class and discuss the connection between these two classes. Finally we bound the metric entropy of bounded variation function class using analysis about Besov class.

## D.1 Definition of Besov Class

Let $\Omega$ be the domain of the function class (which we omit in the definition) and $\|\cdot\|_p$ denote the $\ell_p$ norm. We first define the modulus of smoothness.

**Definition D.1.** For a function $f \in L^p(\Omega)$ where $1 \leq p \leq \infty$, the $r$-th modulus of smoothness is defined by

$$w_{r,p}(f,t) = \sup_{h \in \mathbb{R}^d : \|h\|_2 \leq t} \|\Delta_h^r(f)\|_p, \tag{26}$$

where $\Delta$ is defined as

$$\Delta_h^r(f) := \begin{cases} \sum_{j=0}^r \binom{r}{j} (-1)^{r-j} f(x+jh), & \text{if } x \in \Omega, \ x+rh \in \Omega, \\ 0, & \text{otherwise.} \end{cases} \tag{27}$$

Then the norm of Besov space is defined as below.

**Definition D.2.** For $1 \leq p, q \leq \infty$, $\alpha > 0$, $r := \lceil \alpha \rceil + 1$, define

$$|f|_{B_{p,q}^\alpha} = \begin{cases} \left( \int_{t=0}^\infty (t^{-\alpha} w_{r,p}(f,t))^q \frac{dt}{t} \right)^{\frac{1}{q}}, & q < \infty, \\ \sup_{t>0} t^{-\alpha} w_{r,p}(f,t), & q = \infty. \end{cases} \tag{28}$$

Then the norm of Besov space is defined as:

$$\|f\|_{B_{p,q}^\alpha} = \|f\|_p + |f|_{B_{p,q}^\alpha}. \tag{29}$$

Finally, a function $f$ is in the Besov class $B_{p,q}^\alpha$ if $\|f\|_{B_{p,q}^\alpha}$ is finite. For more discussions and properties of Besov class, we refer interesting readers to Edmunds and Triebel [1996].

## D.2 Definition of Bounded Variation Class and the Connection

For the same domain $\Omega$, recall that the bounded variation function class is defined as

$$\mathrm{BV}^{(1)}(B, C_n) := \left\{ f : \Omega \to \mathbb{R} \ \middle| \ \max_x |f(x)| \leq B, \int_{-x_{\max}}^{x_{\max}} |f''(x)| dx \leq C_n \right\}. \tag{30}$$

According to DeVore and Lorentz [1993], bounded total variation class is closely connected to the Besov class. Specifically, for any constant $B, C_n$, it holds that

$$\mathrm{BV}^{(1)}(B, C_n) \subset B_{1,\infty}^2. \tag{31}$$

## D.3 Metric Entropy of Bounded Total Variation Function Class

Now we bound the complexity of $\mathrm{BV}^{(1)}(1,1)$, which will be helpful for bounding the complexity of $\mathrm{BV}^{(1)}(B, C_n)$ in the future. We first define the metric entropy of a metric space $(\mathbb{T}, \rho)$.

**Definition D.3.** For a set $\mathbb{T}$ with a corresponding metric $\rho(\cdot, \cdot)$, let $N(\epsilon, \mathbb{T}, \rho)$ denote the $\epsilon$-covering number of $\mathbb{T}$ under metric $\rho$. Then the metric entropy of $\mathbb{T}$ with respect to $\rho$ is $\log N(\epsilon, \mathbb{T}, \rho)$.

More details and examples about covering and metric entropy can be found in Chapter 5 of Wainwright [2019]. Next we bound the metric entropy of a bounded subset of $BV(1)$. Note that the $\ell_\infty$ metric over domain $\Omega$ is $\rho_\infty(f,g) = \sup_{x \in \Omega} |f(x) - g(x)|$, which we denote by $\|\cdot\|_\infty$ for short.

**Lemma D.4.** *Assume the set* $\mathbb{T}_1 = \left\{ f : [-1,1] \to \mathbb{R} \;\middle|\; \int_{-1}^{1} |f''(x)| dx \leq 1, \; |f(x)| \leq 1 \right\}$ *and the metric is the* $\ell_\infty$ *distance* $\| \cdot \|_\infty$*, then there exists a universal constant* $C_1 > 0$ *such that for any* $\epsilon > 0$*, the metric entropy of* $(\mathbb{T}_1, \| \cdot \|_\infty)$ *satisfies*

$$\log N(\epsilon, \mathbb{T}_1, \| \cdot \|_\infty) \leq C_1 \epsilon^{-\frac{1}{2}}. \tag{32}$$

*Proof of Lemma D.4.* First of all, the domain $\Omega = [-1,1]$ is a bounded set in $\mathbb{R}$. Moreover, according to DeVore and Lorentz [1993], we have $BV(1) \subset B_{1,\infty}^2$. Therefore, $\mathbb{T}_1$ is a bounded subset of $B_{1,\infty}^2(\Omega)$ with both $B_{1,\infty}^2$ and $\ell_\infty$ norm bounded by a universal constant.

Therefore, the metric space $(\mathbb{T}_1, \| \cdot \|_\infty)$ satisfies the assumptions in (the second point of) Corollary 2 in Nickl and Pötscher [2007] with $r = \infty$, $d = 1$, $s = 2$. Then combining the conclusion of Corollary 2 in Nickl and Pötscher [2007] and the fact that the metric entropy is upper bounded by bracketing metric entropy (Definition 4 in Nickl and Pötscher [2007]), we finish the proof. $\quad\square$

**Remark D.5.** For our purpose of bounding the metric entropy of $BV(1)$, we only consider the case where $\Omega$ is bounded. For more results regarding the metric entropy of (weighted) Besov space, please refer to Nickl and Pötscher [2007].

# E   Calculation of Gradient and Hessian Matrix

In this section, we calculate the gradient and Hessian matrix of $f_\theta(x)$ with respect to $\theta$. Recall that $f_\theta(x) = \sum_{i=1}^{k} w_i^{(2)} \phi\left(w_i^{(1)} x + b_i^{(1)}\right) + b^{(2)}$ where $\phi(x) = \max\{x, 0\}$. We denote $\theta = (w_1^{(1)}, \cdots, w_k^{(1)}, b_1^{(1)}, \cdots, b_k^{(1)}, w_1^{(2)}, \cdots, w_k^{(2)}, b^{(2)})^T$.

## E.1   Calculation of Gradient

According to direct calculation, for a given $x \in [-x_{\max}, x_{\max}]$ we have

$$\begin{cases} \nabla_{w_i^{(1)}} f_\theta(x) = x w_i^{(2)} \mathbb{1}\left(w_i^{(1)} x + b_i^{(1)} > 0\right), \;\; \forall\, i \in [k] \\ \nabla_{b_i^{(1)}} f_\theta(x) = w_i^{(2)} \mathbb{1}\left(w_i^{(1)} x + b_i^{(1)} > 0\right), \;\; \forall\, i \in [k] \\ \nabla_{w_i^{(2)}} f_\theta(x) = \phi\left(w_i^{(1)} x + b_i^{(1)}\right) = \left(w_i^{(1)} x + b_i^{(1)}\right) \mathbb{1}\left(w_i^{(1)} x + b_i^{(1)} > 0\right), \;\; \forall\, i \in [k] \\ \nabla_{b^{(2)}} f_\theta(x) = 1 \end{cases} \tag{33}$$

## E.2   Calculation of the Hessian Matrix

In this part, we calculate $\nabla_\theta^2 f_\theta(x)$ for a given $x \in [-x_{\max}, x_{\max}]$. Below we calculate $\frac{\partial^2 f_\theta(x)}{\partial \theta_i \partial \theta_j}$.

First of all, if $\theta_i = b^{(2)}$ or $\theta_j = b^{(2)}$, $\frac{\partial^2 f_\theta(x)}{\partial \theta_i \partial \theta_j} = 0$. Then it remains to calculate $\frac{\partial^2 f_\theta(x)}{\partial \theta_i \partial \theta'_j}$ where $i, j \in [k]$ and $\theta, \theta' \in \{w^{(1)}, b^{(1)}, w^{(2)}\}$. It is obvious that if $i \neq j$, $\frac{\partial^2 f_\theta(x)}{\partial \theta_i \partial \theta'_j} = 0$. Therefore, we only calculate the case when $j = i$. Let $\delta$ denote the Dirac function, it holds that:

$$\begin{cases} \frac{\partial^2 f_\theta(x)}{\partial w_i^{(1)} \partial w_i^{(1)}} = w_i^{(2)} x^2 \delta\left(w_i^{(1)} x + b_i^{(1)}\right), \;\; \forall\, i \in [k] \\ \frac{\partial^2 f_\theta(x)}{\partial w_i^{(1)} \partial b_i^{(1)}} = \frac{\partial^2 f_\theta(x)}{\partial b_i^{(1)} \partial w_i^{(1)}} = x w_i^{(2)} \delta\left(w_i^{(1)} x + b_i^{(1)}\right), \;\; \forall\, i \in [k] \\ \frac{\partial^2 f_\theta(x)}{\partial b_i^{(1)} \partial b_i^{(1)}} = w_i^{(2)} \delta\left(w_i^{(1)} x + b_i^{(1)}\right), \;\; \forall\, i \in [k] \\ \frac{\partial^2 f_\theta(x)}{\partial w_i^{(2)} \partial w_i^{(2)}} = 0, \;\; \forall\, i \in [k] \\ \frac{\partial^2 f_\theta(x)}{\partial w_i^{(1)} \partial w_i^{(2)}} = \frac{\partial^2 f_\theta(x)}{\partial w_i^{(2)} \partial w_i^{(1)}} = x \mathbb{1}\left(w_i^{(1)} x + b_i^{(1)} > 0\right), \;\; \forall\, i \in [k] \\ \frac{\partial^2 f_\theta(x)}{\partial b_i^{(1)} \partial w_i^{(2)}} = \frac{\partial^2 f_\theta(x)}{\partial w_i^{(2)} \partial b_i^{(1)}} = \mathbb{1}\left(w_i^{(1)} x + b_i^{(1)} > 0\right), \;\; \forall\, i \in [k] \end{cases} \tag{34}$$

The gradient is generally not well-defined according to the existence of the Dirac function. However, under the assumption that $f_\theta$ is twice differentiable with respect to $\theta$ (*i.e.* the knots of $f$ do not coincide with $x$), all the Dirac functions take the value 0. In this case,

$$
\begin{cases}
\frac{\partial^2 f_\theta(x)}{\partial w_i^{(1)} \partial w_i^{(1)}} = 0, & \forall\, i \in [k] \\
\frac{\partial^2 f_\theta(x)}{\partial w_i^{(1)} \partial b_i^{(1)}} = \frac{\partial^2 f_\theta(x)}{\partial b_i^{(1)} \partial w_i^{(1)}} = 0, & \forall\, i \in [k] \\
\frac{\partial^2 f_\theta(x)}{\partial b_i^{(1)} \partial b_i^{(1)}} = 0, & \forall\, i \in [k]
\end{cases}
\tag{35}
$$

### E.3  Upper Bound of Operator Norm

In this part, we upper bound the operator norm of the Hessian matrix. Equivalently, we upper bound $\left|v^T \nabla^2 f_\theta(x) v\right|$ under the constraint that $\|v\|_2 = 1$. We have the following lemma.

**Lemma E.1.** *Assume that $f_\theta(x)$ is twice differentiable with respect to $\theta$ and $x \in [-x_{\max}, x_{\max}]$, for any $v$ such that $\|v\|_2 = 1$, it holds that*

$$
\left|v^T \nabla^2 f_\theta(x) v\right| \le 2 \max\{x_{\max}, 1\}.
\tag{36}
$$

*Proof of Lemma E.1.* Assume $v = (\alpha_1, \cdots, \alpha_k, \beta_1, \cdots, \beta_k, \gamma_1, \cdots, \gamma_k, \iota)^T \in \mathbb{R}^{3k+1}$ such that $\sum_{i=1}^k (\alpha_i^2 + \beta_i^2 + \gamma_i^2) + \iota^2 = 1$. Note that the Hessian matrix $\nabla_\theta^2 f_\theta(x)$ follows the structure:

$$
\nabla_\theta^2 f_\theta(x) = \begin{pmatrix}
A_{w^{(1)}w^{(1)}} & A_{w^{(1)}b^{(1)}} & A_{w^{(1)}w^{(2)}} & A_{w^{(1)}b^{(2)}} \\
A_{b^{(1)}w^{(1)}} & A_{b^{(1)}b^{(1)}} & A_{b^{(1)}w^{(2)}} & A_{b^{(1)}b^{(2)}} \\
A_{w^{(2)}w^{(1)}} & A_{w^{(2)}b^{(1)}} & A_{w^{(2)}w^{(2)}} & A_{w^{(2)}b^{(2)}} \\
A_{b^{(2)}w^{(1)}} & A_{b^{(2)}b^{(1)}} & A_{b^{(2)}w^{(2)}} & A_{b^{(2)}b^{(2)}}
\end{pmatrix}
\tag{37}
$$

where $A_{w^{(1)}w^{(1)}}, A_{w^{(1)}b^{(1)}}, A_{b^{(1)}w^{(1)}}, A_{b^{(1)}b^{(1)}}, A_{w^{(2)}w^{(2)}} \in \mathbb{R}^{k \times k}$, $A_{w^{(1)}b^{(2)}}, A_{b^{(1)}b^{(2)}}, A_{w^{(2)}b^{(2)}} \in \mathbb{R}^{k \times 1}$, $A_{b^{(2)}w^{(1)}}, A_{b^{(2)}b^{(1)}}, A_{b^{(2)}w^{(2)}} \in \mathbb{R}^{1 \times k}$ and $A_{b^{(2)}b^{(2)}} \in \mathbb{R}$ are all zero matrices. Meanwhile, $A_{w^{(1)}w^{(2)}}, A_{b^{(1)}w^{(2)}}, A_{w^{(2)}w^{(1)}}, A_{w^{(2)}b^{(1)}} \in \mathbb{R}^{k \times k}$ are all diagonal matrices whose non-zero elements are between $[-\max\{x_{\max}, 1\}, \max\{x_{\max}, 1\}]$. Therefore, it holds that:

$$
\begin{aligned}
\left|v^T \nabla^2 f_\theta(x) v\right| &\le 2 \max\{x_{\max}, 1\} \sum_{i=1}^k \left(|\alpha_i \gamma_i| + |\beta_i \gamma_i|\right) \\
&\le 2 \max\{x_{\max}, 1\} \left( \sqrt{\sum_{i=1}^k \alpha_i^2 \cdot \sum_{i=1}^k \gamma_i^2} + \sqrt{\sum_{i=1}^k \beta_i^2 \cdot \sum_{i=1}^k \gamma_i^2} \right) \\
&\le 2 \max\{x_{\max}, 1\},
\end{aligned}
\tag{38}
$$

where the second inequality holds because of Cauchy-Schwarz inequality. The last inequality results from $x(1-x) \le \frac{1}{4}$. $\qquad\square$

## F  Proof for the Counter-Example (Theorem 3.1)

**Theorem F.1** (Restate Theorem 3.1). *For the counter-example, with probability $1 - \delta$, for any interpolating function $f$,*

$$
\int_{-x_{\max}}^{x_{\max}} |f''(x)| g(x) dx = \Omega\left( \sigma n \left[ n - 24 \log\left(\frac{1}{\delta}\right) \right] \right),
\tag{39}
$$

*where the randomness comes from the noises $\{\epsilon_i\}$. Under this high-probability event, when $n \ge \Omega\left( \sqrt{\frac{1}{\sigma\eta}} \log\left(\frac{1}{\delta}\right) \right)$, any stable solution $f$ for GD with step size $\eta$ will not interpolate the data, i.e.*

$$
\mathcal{F}(\eta, 0, \mathcal{D}) \cap \{f \mid f(x_i) = y_i, \ \forall\, i \in [n]\} = \emptyset.
\tag{40}
$$

*Proof of Theorem F.1.* Consider any three consecutive points $x_j, x_{j+1}, x_{j+2}$ where $j \in [n-2]$, note that their corresponding $y$'s are $y_j, y_{j+1}, y_{j+2}$ which are i.i.d. Gaussian random variables $\mathcal{N}(0, \sigma^2)$. Then according to Mean Value Theorem, there exists $a \in [x_j, x_{j+1}]$ and $b \in [x_{j+1}, x_{j+2}]$ such that

$$f'(a) = \frac{y_{j+1} - y_j}{x_{j+1} - x_j} = \frac{n-1}{2x_{\max}}(y_{j+1} - y_j), \quad f'(b) = \frac{y_{j+2} - y_{j+1}}{x_{j+2} - x_{j+1}} = \frac{n-1}{2x_{\max}}(y_{j+2} - y_{j+1}). \quad (41)$$

Therefore, it holds that

$$\int_{x_j}^{x_{j+2}} |f''(x)|dx \geq |f'(b) - f'(a)| = \frac{n-1}{2x_{\max}}|y_{j+2} - 2y_{j+1} + y_j| \sim \frac{n-1}{2x_{\max}} \cdot |\mathcal{N}(0, 6\sigma^2)|, \quad (42)$$

where the last equation means that $y_{j+2} - 2y_{j+1} + y_j$ follows the distribution $\mathcal{N}(0, 6\sigma^2)$.

We focus on the interval in the middle. For any $x \in [x_{n/4}, x_{3n/4}]$, we have

$$\mathbb{P}^2(X < x) \geq \frac{1}{16}, \quad \mathbb{E}[x - X | X < x] \geq \frac{x_{\max}}{4},$$
$$\mathbb{P}^2(X > x) \geq \frac{1}{16}, \quad \mathbb{E}[X - x | X > x] \geq \frac{x_{\max}}{4}. \quad (43)$$

Together with the definition of $g(x)$ (6), we have for any $x \in [x_{n/4}, x_{3n/4}]$, $g(x) \geq \frac{x_{\max}}{64}$. Therefore, for any interpolating solutions $f$, it holds that

$$\int_{-x_{\max}}^{x_{\max}} |f''(x)|g(x)dx \geq \frac{x_{\max}}{64} \int_{x_{n/4}}^{x_{3n/4}} |f''(x)|dx \geq \frac{x_{\max}}{64} \cdot \frac{n-1}{2x_{\max}} \sum_{i=1}^{n/6} |G_i|, \quad (44)$$

where $G_i$'s are i.i.d samples from $\mathcal{N}(0, 6\sigma^2)$.

Assume the median of $|\mathcal{N}(0, 1)|$ is $c > 0$, which is a universal constant. For any $i \in [\frac{n}{6}]$, define

$$H_i = \begin{cases} \sqrt{6}c\sigma, & \text{if } |G_i| \geq \sqrt{6}c\sigma, \\ 0, & \text{otherwise.} \end{cases} \quad (45)$$

Then we have $H_i = \sqrt{6}c\sigma$ with probability $\frac{1}{2}$. In addition, $|G_i| \geq H_i$. According to Lemma K.1, with probability at least $1 - \delta$,

$$\int_{-x_{\max}}^{x_{\max}} |f''(x)|g(x)dx \geq \frac{n-1}{128} \sum_{i=1}^{n/6} H_i \geq \frac{n-1}{128} \cdot \sqrt{6}c\sigma \cdot \left( \frac{n}{24} - \log\left( \frac{1}{\delta} \right) \right) = c'\sigma n \left( n - 24\log\left( \frac{1}{\delta} \right) \right), \quad (46)$$

for some universal constant $c'$.

Together with the conclusion in Theorem C.2, for any interpolating and stable solution $f$, with probability $1 - \delta$,

$$\frac{1}{\eta} - \frac{1}{2} \geq c'\sigma n \left( n - 24\log\left( \frac{1}{\delta} \right) \right), \quad (47)$$

which does not hold when $n \geq \Omega\left( \sqrt{\frac{1}{\sigma\eta}} \log\left( \frac{1}{\delta} \right) \right)$. $\square$

# G  Proof for the $\text{TV}^{(1)}$ Bound (Theorem 4.1)

We begin by calculating the gradient of empirical loss $\mathcal{L}$ at $\theta$:

$$\nabla_\theta \mathcal{L}(\theta) = \frac{1}{n} \sum_{i=1}^{n} (f_\theta(x_i) - y_i)\nabla_\theta f_\theta(x_i), \quad (48)$$

where the detailed calculation of $\nabla_\theta f_\theta(x)$ can be found in Appendix E. Then the Hessian is given by

$$\nabla_\theta^2 \mathcal{L}(\theta) = \frac{1}{n} \sum_{i=1}^{n} (\nabla_\theta f_\theta(x_i))(\nabla_\theta f_\theta(x_i))^T + \frac{1}{n} \sum_{i=1}^{n} (f_\theta(x_i) - y_i)\nabla_\theta^2 f(x_i), \quad (49)$$

where $\nabla_\theta^2 f(x)$ is calculated in Appendix E. Let $v$ denote the unit eigenvector ($\|v\|_2 = 1$) of $\frac{1}{n}\sum_{i=1}^n (\nabla_\theta f_\theta(x_i))(\nabla_\theta f_\theta(x_i))^T$ with respect to the largest eigenvalue, it holds that

$$\lambda_{\max}(\nabla_\theta^2 \mathcal{L}(\theta)) \geq v^T \nabla_\theta^2 \mathcal{L}(\theta) v$$

$$= \underbrace{\lambda_{\max}\left(\frac{1}{n}\sum_{i=1}^n (\nabla_\theta f_\theta(x_i))(\nabla_\theta f_\theta(x_i))^T\right)}_{(\star)} + \underbrace{\frac{1}{n}\sum_{i=1}^n (f_\theta(x_i) - y_i) v^T \nabla_\theta^2 f(x_i) v}_{(\#)}$$

$$= \underbrace{\lambda_{\max}\left(\frac{1}{n}\sum_{i=1}^n (\nabla_\theta f_\theta(x_i))(\nabla_\theta f_\theta(x_i))^T\right)}_{(\star)} + \underbrace{\frac{1}{n}\sum_{i=1}^n (f_0(x_i) - y_i) v^T \nabla_\theta^2 f(x_i) v}_{(i)} \qquad (50)$$

$$+ \underbrace{\frac{1}{n}\sum_{i=1}^n (f_\theta(x_i) - f_0(x_i)) v^T \nabla_\theta^2 f(x_i) v}_{(ii)}.$$

For term $(\star)$, we connect the maximal eigenvalue at $\theta$ to the (weighted) $\mathrm{TV}^{(1)}$ norm of the corresponding $f = f_\theta$. Let the weight function $g(x)$ be defined as (6), then the lemma below holds.

**Lemma G.1.** *Assume $\mathcal{L}$ is twice differentiable at $\theta$ and the corresponding function of $\theta$ is $f$, then*

$$\lambda_{\max}\left(\frac{1}{n}\sum_{i=1}^n (\nabla_\theta f_\theta(x_i))(\nabla_\theta f_\theta(x_i))^T\right) \geq 1 + 2\int_{-x_{\max}}^{x_{\max}} |f''(x)| g(x) dx, \qquad (51)$$

*where $g(x)$ is defined as (6).*

*Proof of Lemma G.1.* The proof of Lemma 4 in Mulayoff et al. [2021] directly proves the result for $\lambda_{\max}\left(\frac{1}{n}\sum_{i=1}^n (\nabla_\theta f_\theta(x_i))(\nabla_\theta f_\theta(x_i))^T\right)$, which is the $\lambda_{\max}(\nabla_\theta^2 \mathcal{L})$ in Lemma 4 of Mulayoff et al. [2021] when $f$ is an interpolating solution. $\square$

For the first inequality of Theorem 4.1, we directly handle the term $(\#)$ as below.

$$|(\#)| \leq \sqrt{\frac{1}{n}\sum_{i=1}^n (f_\theta(x_i) - y_i)^2} \cdot \sqrt{\frac{1}{n}\sum_{i=1}^n (v^T \nabla_\theta^2 f(x_i) v)^2} \leq 2x_{\max}\sqrt{2\mathcal{L}(\theta)}, \qquad (52)$$

where the first inequality results from Cauchy-Schwarz inequality. The second inequality is because of the uniform upper bound of $v^T \nabla_\theta^2 f(x_i) v$ (Lemma E.1).

For the second inequality of Theorem 4.1, we bound the two terms (i) and (ii). We begin with $|(i)| = \left|\frac{1}{n}\sum_{i=1}^n v^T \nabla_\theta^2 f(x_i) v \cdot \epsilon_i\right|$, which is a weighted sum of noises $\{\epsilon_i\}$.

**Lemma G.2.** *Assume $\epsilon_i$'s are independently sampled from $\mathcal{N}(0, \sigma^2)$ for some $\sigma > 0$, with probability at least $1 - \delta$, uniformly over all $\theta, v$ such that $\mathcal{L}$ is twice differentiable at $\theta$ and $\|v\|_2 = 1$,*

$$\left|\frac{1}{n}\sum_{i=1}^n v^T \nabla_\theta^2 f(x_i) v \cdot \epsilon_i\right| \leq \sigma \max\{x_{\max}, 1\} \cdot \min\left\{4\sqrt{\log\left(\frac{4n}{\delta}\right)}, 14\sqrt{\frac{k\log\left(\frac{13n}{\delta}\right)}{n}}\right\}. \qquad (53)$$

*Proof of Lemma G.2.* For the first part, according to Lemma E.1, we have

$$\left|\frac{1}{n}\sum_{i=1}^n v^T \nabla_\theta^2 f(x_i) v \cdot \epsilon_i\right| \leq 2\max\{x_{\max}, 1\} \cdot \max_i\{|\epsilon_i|\}. \qquad (54)$$

Since $\epsilon_i$'s are independently sampled from Gaussian distribution $\mathcal{N}(0, \sigma^2)$, according to concentration of Gaussian distribution and a union bound, with probability $1 - \frac{\delta}{2}$, it holds that:

$$\max_i\{|\epsilon_i|\} \leq 2\sigma\sqrt{\log\left(\frac{4n}{\delta}\right)}. \qquad (55)$$

Under this high-probability event, we have

$$|(i)| = \left| \frac{1}{n} \sum_{i=1}^{n} v^T \nabla_\theta^2 f(x_i) v \cdot \epsilon_i \right| \le 4\sigma \max\{x_{\max}, 1\} \sqrt{\log\left(\frac{4n}{\delta}\right)}. \tag{56}$$

For the second part, we bound the complexity of $\left\{ v^T \nabla_\theta^2 f(x_i) v \right\}_{i=1}^{n}$. Notice that $\theta$ is a function of the dataset, thus not independent to $\epsilon_i$. Also, $v$ is a function of the dataset and $\theta$. Our strategy is to apply an $\epsilon$-net argument for both $v$ and $\nabla_\theta^2 f(x_i)$ for $i = 1, \cdots, n$.

We begin with considering the possibilities of $\{\nabla_\theta^2 f(x_i)\}_{i=1}^{n}$. According to the detailed form of $\nabla_\theta^2 f_\theta(x)$ in Appendix E, we have the set $\{\nabla_\theta^2 f(x_i)\}_{i=1}^{n}$ is fully determined by $\left\{ \mathbb{1}\left( w_j^{(1)} x_i + b_j^{(1)} > 0 \right) \right\}_{i,j=1,1}^{n,k}$. Therefore it suffices to cover all the possibilities of $\left\{ \mathbb{1}\left( w_j^{(1)} x_i + b_j^{(1)} > 0 \right) \right\}_{i=1}^{n}$ for all $j \in [k]$. Without loss of generality, we can assume that $x_1 < x_2 < \cdots < x_n$, and then $\left\{ w_j^{(1)} x_i + b_j^{(1)} \right\}_{i=1}^{n}$ is also monotonic, which implies that there are $2(n+1)$ possibilities of $\left\{ \mathbb{1}\left( w_j^{(1)} x_i + b_j^{(1)} > 0 \right) \right\}_{i=1}^{n}$ in total. As a result, the product space $\left\{ \mathbb{1}\left( w_j^{(1)} x_i + b_j^{(1)} > 0 \right) \right\}_{i,j=1,1}^{n,k}$ (and also $\{\nabla_\theta^2 f(x_i)\}_{i=1}^{n}$) has $N_1 = (2n+2)^k$ possibilities.

For a fixed matrix $M = \nabla_\theta^2 f(x)$ for some $\theta, x$ and $v, v'$ such that $\|v\|_2 \le 1, \|v'\|_2 \le 1, \|v - v'\|_2 \le \epsilon$ with $\epsilon \in (0, 1)$, it holds that

$$
\begin{aligned}
\left| v^T M v - (v')^T M v' \right| &\le 2 \left| (v - v')^T M v' \right| + \left| (v - v')^T M (v - v') \right| \\
&\le 2\|v - v'\|_2 \|M\|_2 \|v'\|_2 + \|v - v'\|_2 \|M\|_2 \|v - v'\|_2 \\
&\le 4 \max\{x_{\max}, 1\}\epsilon + 2 \max\{x_{\max}, 1\}\epsilon^2 \\
&\le 6 \max\{x_{\max}, 1\}\epsilon,
\end{aligned}
\tag{57}
$$

where the third inequality results from the upper bound of operator norm of $M$ (Lemma E.1). Therefore, the exact covering set of $\{\nabla_\theta^2 f(x_i)\}_{i=1}^{n}$ and an $\frac{\epsilon}{6\max\{x_{\max},1\}}$-covering set of the unit Euclidean Ball with dimension $3k + 1$ (which is exactly the domain of $v$) together provides an $\epsilon$-cover of $\left\{ \left( v^T \nabla_\theta^2 f(x_i) v \right)_{i=1}^{n} \right\}$ with respect to $\|\cdot\|_\infty$. Meanwhile, an $\frac{\epsilon}{6\max\{x_{\max},1\}}$-cover with respect to $\|\cdot\|_2$ of the unit Euclidean Ball with dimension $3k + 1$ has cardinality bounded by $N_2 = \left(1 + \frac{12\max\{x_{\max},1\}}{\epsilon}\right)^{3k+1}$ according to Lemma K.2. Combining the two covering arguments, the $\epsilon$-covering set of $\left\{ \left( v^T \nabla_\theta^2 f(x_i) v \right)_{i=1}^{n} \right\}$ with respect to $\|\cdot\|_\infty$ has cardinality $N_\epsilon$ satisfying:

$$\log N_\epsilon \le \log N_1 + \log N_2 \le 4k \log\left(\frac{13n \max\{x_{\max}, 1\}}{\epsilon}\right). \tag{58}$$

Consider a fixed stream $(a_i)_{i=1}^{n}$ in the covering set, we have $|a_i| \le 2 \max\{x_{\max}, 1\}$ for all $i \in [n]$. Then according to the concentration result of Gaussian distribution, with probability $1 - \delta$,

$$\left| \frac{1}{n} \sum_{i=1}^{n} a_i \epsilon_i \right| \le 4\sigma \max\{x_{\max}, 1\} \sqrt{\frac{\log(2/\delta)}{n}}. \tag{59}$$

With a union bound over the $\epsilon$-covering set of $\left\{ \left( v^T \nabla_\theta^2 f(x_i) v \right)_{i=1}^{n} \right\}$ and conditioned on the high probability event of (55), with probability $1 - \delta$, uniformly over all possible $\|v\|_2 = 1$ and $\theta$,

$$
\begin{aligned}
\left| \frac{1}{n} \sum_{i=1}^{n} v^T \nabla_\theta^2 f(x_i) v \cdot \epsilon_i \right| &\le 2\sigma\epsilon \sqrt{\log\left(\frac{4n}{\delta}\right)} + 4\sigma \max\{x_{\max}, 1\} \sqrt{\frac{\log(4N_\epsilon/\delta)}{n}} \\
&\le 2\sigma\epsilon \sqrt{\log\left(\frac{4n}{\delta}\right)} + 4\sigma \max\{x_{\max}, 1\} \sqrt{\frac{4k \log\left(\frac{13n \max\{x_{\max},1\}}{\epsilon\delta}\right)}{n}}.
\end{aligned}
\tag{60}
$$

Finally, by choosing $\epsilon = \frac{\max\{x_{\max}, 1\}}{\sqrt{n}}$, we have

$$\left| \frac{1}{n} \sum_{i=1}^{n} v^T \nabla_\theta^2 f(x_i) v \cdot \epsilon_i \right| \leq 14\sigma \max\{x_{\max}, 1\} \sqrt{\frac{k \log\left(\frac{13n}{\delta}\right)}{n}}. \tag{61}$$

$\square$

**Remark G.3.** According to the Lemma G.2 above, there are two cases. When the neural network is over-parameterized, we can derive a constant upper bound for term (i). Meanwhile, if the number of neurons is smaller than the sample complexity, we can derive a tighter bound for (i) by covering $\left\{ v^T \nabla_\theta^2 f(x_i) v \right\}_{i=1}^{n}$. For $k, n$ such that $k$ is polynomially smaller than $n$ (e.g. $k = n^{1-\alpha}$ for some positive $\alpha$), the term (i) will vanish if $n$ converges to infinity.

Meanwhile, the term (ii) can be bounded by the mean squared error $\mathrm{MSE}(f_\theta)$ using Cauchy-Schwarz inequality and the uniform upper bound of $v^T \nabla_\theta^2 f(x_i) v$ (Lemma E.1).

$$|(ii)| \leq \sqrt{\frac{1}{n} \sum_{i=1}^{n} (f_\theta(x_i) - f_0(x_i))^2} \cdot \sqrt{\frac{1}{n} \sum_{i=1}^{n} (v^T \nabla_\theta^2 f(x_i) v)^2} \leq 2 \max\{x_{\max}, 1\} \sqrt{\mathrm{MSE}(f_\theta)}. \tag{62}$$

Combining the results above, we state the following weighted $\mathrm{TV}^{(1)}$ bound of the learned function. We leave the training loss and mean squared error (MSE) in the bound, which will be handled later.

**Theorem G.4** (Restate Theorem 4.1). *For a function $f = f_\theta$ where the training (square) loss $\mathcal{L}$ is twice differentiable at $\theta$,*

$$\int_{-x_{\max}}^{x_{\max}} |f''(x)| g(x) dx \leq \frac{\lambda_{\max}(\nabla_\theta^2 \mathcal{L}(\theta))}{2} - \frac{1}{2} + x_{\max}\sqrt{2\mathcal{L}(\theta)}, \tag{63}$$

*where $g(x)$ is defined as (6). Moreover, if we assume $y_i = f_0(x_i) + \epsilon_i$ for independent noise $\epsilon_i \sim \mathcal{N}(0, \sigma^2)$, then with probability $1 - \delta$ where the randomness is over the noises $\{\epsilon_i\}$,*

$$\int_{-x_{\max}}^{x_{\max}} |f''(x)| g(x) dx \leq \frac{\lambda_{\max}(\nabla_\theta^2 \mathcal{L}(\theta))}{2} - \frac{1}{2} + \widetilde{O}\left(\sigma x_{\max} \cdot \min\left\{1, \sqrt{\frac{k}{n}}\right\}\right) + x_{\max}\sqrt{\mathrm{MSE}(f)}. \tag{64}$$

*In addition, if $f = f_\theta$ is a* stable solution *of GD with step size $\eta$ on dataset $\mathcal{D}$, i.e., $f_\theta \in \mathcal{F}(\eta, 0, \mathcal{D})$ as in (4), then we can replace $\frac{\lambda_{\max}(\nabla_\theta^2 \mathcal{L}(\theta))}{2}$ with $\frac{1}{\eta}$ in (9) and (10).*

*Proof of Theorem G.4.* The first inequality results from plugging Lemma G.1 and (52) into (50). The second inequality holds by plugging Lemma G.1, Lemma G.2 and inequality (62) into (50). For the instantiation of $\lambda_{\max}(\nabla_\theta^2 \mathcal{L}(\theta))$, the replacement holds due to the definition of $\mathcal{F}(\eta, 0, \mathcal{D})$ in (4). $\square$

## G.1   A Crude Bound for MSE and the Instantiated $\mathrm{TV}^{(1)}$ Bound (Corollary 4.2)

Now we prove a crude upper bound for MSE, which could instantiate a weighted $\mathrm{TV}^{(1)}$ bound.

**Lemma G.5.** *Assume that the function $f$ is optimized, then with probability $1 - \delta$, we have*

$$\mathrm{MSE}(f) = \frac{1}{n} \sum_{i=1}^{n} (f(x_i) - f_0(x_i))^2 \leq 16\sigma^2 \log\left(\frac{2n}{\delta}\right). \tag{65}$$

*Proof of Lemma G.5.* According to the assumption that $f$ is optimized, it holds that

$$\sum_{i=1}^{n} (f(x_i) - y_i)^2 \leq \sum_{i=1}^{n} (f_0(x_i) - y_i)^2 = \sum_{i=1}^{n} \epsilon_i^2. \tag{66}$$

Then since $(a+b)^2 \leq 2a^2 + 2b^2$ always holds (AM-GM inequality), we have

$$
\begin{aligned}
\frac{1}{n} \sum_{i=1}^{n} (f(x_i) - f_0(x_i))^2 &= \frac{1}{n} \sum_{i=1}^{n} (f(x_i) - y_i + y_i - f_0(x_i))^2 \\
&\leq \frac{2}{n} \left[ \sum_{i=1}^{n} (f(x_i) - y_i)^2 + \sum_{i=1}^{n} (f_0(x_i) - y_i)^2 \right] \\
&\leq \frac{4}{n} \sum_{i=1}^{n} \epsilon_i^2 \leq 4 \max_i \epsilon_i^2.
\end{aligned}
\tag{67}
$$

Recall that $\epsilon_i$'s are independently sampled from $\mathcal{N}(0, \sigma^2)$, then according to the concentration of Gaussian distribution and a union bound, with probability $1 - \delta$,

$$
\max_i \epsilon_i^2 \leq 4\sigma^2 \log \left( \frac{2n}{\delta} \right).
\tag{68}
$$

Combining the two results, with probability $1 - \delta$ where the randomness is over the noises $\{\epsilon_i\}$,

$$
\mathrm{MSE}(f) = \frac{1}{n} \sum_{i=1}^{n} (f(x_i) - f_0(x_i))^2 \leq 16\sigma^2 \log \left( \frac{2n}{\delta} \right).
\tag{69}
$$

$\square$

Finally, Corollary 4.2 results from directly plugging Lemma G.5 into Theorem 4.1.

## G.2 An Improved MSE Bound and the Corresponding $\mathrm{TV}^{(1)}$ Bound

In this part, we provide an improved upper bound of the mean squared error $\mathrm{MSE}(f_\theta)$ and also the term (ii). We first make the assumption below that the parameters are from a bounded space.

**Assumption G.6.** There exists some constant $\rho > 0$ such that gradient descent converges to some local minimum $\theta$ with $\|\theta\|_\infty \leq \rho$. In addition, we assume that $\|f_0\|_\infty \leq D$ where $D > 0$ is some universal constant satisfying that $D \leq k\rho^2(x_{\max} + 1)$.

Assumption G.6 ensures that the parameter space is bounded while the ground truth function $f_0$ can be approximated well by some possible output function. The assumption will surely hold for some large enough constants $\rho, D$, which is without loss of generality. In the following analysis, we will replace $D$ with its upper bound $k\rho^2(x_{\max} + 1)$ to reduce the parameters in the logarithmic terms.

To handle the mean squared error $\mathrm{MSE}(f_\theta)$, we begin with an analysis on the complexity of the function class of two-layer ReLU networks with bounded parameters $\mathcal{F}_\rho = \left\{ f : [-x_{\max}, x_{\max}] \to \mathbb{R} \mid f(x) = \sum_{i=1}^{k} w_i^{(2)} \phi\left( w_i^{(1)} x + b_i^{(1)} \right) + b^{(2)}, \|\theta\|_\infty \leq \rho \right\}$. Note that here we assume that the input is from $[-x_{\max}, x_{\max}]$ and there exists an upper bound $\rho > 0$ on the parameter $\theta = (w_1^{(1)}, \cdots, w_k^{(1)}, b_1^{(1)}, \cdots, b_k^{(1)}, w_1^{(2)}, \cdots, w_k^{(2)}, b^{(2)})^T \in \mathbb{R}^{3k+1}$.

**Lemma G.7.** The $\epsilon$-covering number $N_\epsilon$ of function class $\mathcal{F}_\rho$ with respect to $\| \cdot \|_\infty$ satisfies

$$
\log N_\epsilon \leq 4k \log \left( \frac{11 \max\{x_{\max}, 1\} k\rho^2}{\epsilon} \right).
\tag{70}
$$

*Proof of Lemma G.7.* We consider the discrete function class below

$$
\bar{\mathcal{F}}_{\bar{\epsilon}} = \left\{ f : [-x_{\max}, x_{\max}] \to \mathbb{R} \left| \begin{array}{c} f(x) = \sum_{i=1}^{k} \bar{w}_i^{(2)} \phi\left( \bar{w}_i^{(1)} x + \bar{b}_i^{(1)} \right) + \bar{b}^{(2)} \\ \text{s.t. } \theta_j \in \bar{\epsilon} \cdot \mathbb{Z} \cap [-\rho, \rho], \, \forall \, j \in [3k+1] \end{array} \right. \right\},
\tag{71}
$$

where $\theta_j$ is the $j$-th element of $(\bar{w}_1^{(1)}, \cdots, \bar{w}_k^{(1)}, \bar{b}_1^{(1)}, \cdots, \bar{b}_k^{(1)}, \bar{w}_1^{(2)}, \cdots, \bar{w}_k^{(2)}, \bar{b}^{(2)})^T \in \mathbb{R}^{3k+1}$ and $\bar{\epsilon} \cdot \mathbb{Z}$ is the set $\{\bar{\epsilon} \cdot i, \, i \in \mathbb{Z}\}$. Since for each element, the number of choices is bounded by $\frac{2\rho}{\bar{\epsilon}} + 1$, the total cardinality of $\bar{\mathcal{F}}_{\bar{\epsilon}}$ is bounded by $\left( \frac{2\rho}{\bar{\epsilon}} + 1 \right)^{3k+1}$.

For each function $f \in \mathcal{F}_\rho$ with corresponding parameter $\theta$ ($\|\theta\|_\infty \le \rho$), we choose function $\bar{f}$ from $\bar{\mathcal{F}}_{\bar{\epsilon}}$ with corresponding parameter $\bar{\theta}$ such that $|\bar{\theta}_j - \theta_j|$ is minimized for all $j \in [3k+1]$. According to our definition of $\bar{\mathcal{F}}_{\bar{\epsilon}}$, we have for all $j \in [3k+1]$, $|\bar{\theta}_j - \theta_j| \le \bar{\epsilon}$. Therefore, it holds that

$$
\begin{aligned}
\|f - \bar{f}\|_\infty \le & \sum_{i=1}^k \left\| w_i^{(2)} \phi\left(w_i^{(1)} x + b_i^{(1)}\right) - \bar{w}_i^{(2)} \phi\left(\bar{w}_i^{(1)} x + \bar{b}_i^{(1)}\right) \right\|_\infty + \left| b^{(2)} - \bar{b}^{(2)} \right| \\
\le & \sum_{i=1}^k \left\| w_i^{(2)} \phi\left(w_i^{(1)} x + b_i^{(1)}\right) - \bar{w}_i^{(2)} \phi\left(w_i^{(1)} x + b_i^{(1)}\right) \right\|_\infty \\
& + \sum_{i=1}^k \left\| \bar{w}_i^{(2)} \phi\left(w_i^{(1)} x + b_i^{(1)}\right) - \bar{w}_i^{(2)} \phi\left(\bar{w}_i^{(1)} x + \bar{b}_i^{(1)}\right) \right\|_\infty + \bar{\epsilon} \\
\le & k\bar{\epsilon} \cdot \rho(x_{\max} + 1) + k\rho(x_{\max} + 1)\bar{\epsilon} + \bar{\epsilon} \\
\le & 5 \max\{x_{\max}, 1\} k\rho\bar{\epsilon} \le \epsilon,
\end{aligned}
\tag{72}
$$

where the last inequality is by choosing $\bar{\epsilon} = \frac{\epsilon}{5 \max\{x_{\max}, 1\}k\rho}$.

Therefore, the $\epsilon$-covering number $N_\epsilon$ of function class $\mathcal{F}_\rho$ with respect to $\|\cdot\|_\infty$ satisfies

$$
N_\epsilon \le \left(1 + \frac{10 \max\{x_{\max}, 1\}k\rho^2}{\epsilon}\right)^{3k+1},
\tag{73}
$$

which implies that

$$
\log N_\epsilon \le 4k \log\left(\frac{11 \max\{x_{\max}, 1\}k\rho^2}{\epsilon}\right).
\tag{74}
$$

$\square$

Now we provide an upper bound of $\mathrm{MSE}(f_\theta)$ under the assumption that $\theta$ is optimized, which means that the empirical error of $f_\theta$ is smaller than that of $f_0$, *i.e.*

$$
\frac{1}{2n} \sum_{i=1}^n \left(f_\theta(x_i) - y_i\right)^2 \le \frac{1}{2n} \sum_{i=1}^n \left(f_0(x_i) - y_i\right)^2.
\tag{75}
$$

Below we state the improved upper bound of mean squared error under such assumptions.

**Lemma G.8.** *Assume $\epsilon_i$'s are independently sampled from $\mathcal{N}(0, \sigma^2)$ for some $\sigma > 0$, if Assumption G.6 holds and $f_\theta \in \mathcal{F}_\rho$ is optimized, then with probability at least $1 - \delta$, it holds that*

$$
\mathrm{MSE}(f_\theta) \le O\left(\frac{\sigma^2 k \log\left(\frac{\max\{x_{\max}, 1\}kn\rho}{\delta}\right)}{n}\right).
\tag{76}
$$

*Proof of Lemma G.8.* Since $\frac{1}{2n} \sum_{i=1}^n \left(f_\theta(x_i) - y_i\right)^2 \le \frac{1}{2n} \sum_{i=1}^n \left(f_0(x_i) - y_i\right)^2$, we have

$$
\frac{1}{2}\mathrm{MSE}(f_\theta) = \frac{1}{2n} \sum_{i=1}^n \left(f_\theta(x_i) - f_0(x_i)\right)^2 \le \frac{1}{n} \sum_{i=1}^n \epsilon_i \left(f_\theta(x_i) - f_0(x_i)\right).
\tag{77}
$$

For the optimized function $f_\theta$, we choose a function $\bar{f}$ from the $\epsilon$-covering set in Lemma G.7 such that $\|\bar{f} - f_\theta\|_\infty \le \epsilon$. Due to identical analysis as (55), with probability $1 - \frac{\delta}{2}$, $\max_i\{|\epsilon_i|\} \le 2\sigma\sqrt{\log\left(\frac{4n}{\delta}\right)}$. Under this high-probability event, it holds that

$$
\frac{1}{n} \sum_{i=1}^n \epsilon_i \left(f_\theta(x_i) - f_0(x_i)\right) \le \frac{1}{n} \sum_{i=1}^n \epsilon_i \left(\bar{f}(x_i) - f_0(x_i)\right) + 2\sigma\epsilon\sqrt{\log\left(\frac{4n}{\delta}\right)}.
\tag{78}
$$

According to Lemma G.7, the $\epsilon$-covering number $N_\epsilon$ of function class $\mathcal{F}_\rho$ with respect to $\|\cdot\|_\infty$ satisfies $\log N_\epsilon \le 4k \log\left(\frac{11 \max\{x_{\max}, 1\}k\rho^2}{\epsilon}\right)$. Due to the concentration of Gaussian distribution and

a union bound over the covering set, with probability $1 - \frac{\delta}{2}$, for all $\bar{f}$ in the covering set,

$$\sum_{i=1}^{n} \epsilon_i \left( \bar{f}(x_i) - f_0(x_i) \right) \leq 2\sigma \sqrt{\sum_{i=1}^{n} \left( \bar{f}(x_i) - f_0(x_i) \right)^2 \cdot \log \left( \frac{4N_\epsilon}{\delta} \right)}. \tag{79}$$

Combining the two high-probability events, with probability $1 - \delta$, we have

$$\frac{1}{2}\text{MSE}(f_\theta) \leq \frac{1}{n} \sum_{i=1}^{n} \epsilon_i \left( \bar{f}(x_i) - f_0(x_i) \right) + 2\sigma\epsilon \sqrt{\log \left( \frac{4n}{\delta} \right)}$$

$$\leq \frac{2\sigma}{n} \sqrt{\sum_{i=1}^{n} \left( \bar{f}(x_i) - f_0(x_i) \right)^2 \cdot \log \left( \frac{4N_\epsilon}{\delta} \right)} + 2\sigma\epsilon \sqrt{\log \left( \frac{4n}{\delta} \right)}$$

$$\leq \frac{2\sigma}{n} \sqrt{\sum_{i=1}^{n} \left( \bar{f}(x_i) - f_0(x_i) \right)^2 \cdot 4k \log \left( \frac{11 \max\{x_{\max}, 1\} k\rho^2}{\epsilon\delta} \right)} + 2\sigma\epsilon \sqrt{\log \left( \frac{4n}{\delta} \right)}$$

$$\leq \frac{2\sigma}{n} \sqrt{\left[ \sum_{i=1}^{n} \left( f_\theta(x_i) - f_0(x_i) \right)^2 + 10nk\rho^2 \max\{x_{\max}, 1\}\epsilon \right] \cdot 4k \log \left( \frac{11 \max\{x_{\max}, 1\} k\rho^2}{\epsilon\delta} \right)}$$

$$+ 2\sigma\epsilon \sqrt{\log \left( \frac{4n}{\delta} \right)}$$

$$\leq O \left( \sigma \sqrt{k \log \left( \frac{\max\{x_{\max}, 1\} k\rho}{\epsilon\delta} \right) \cdot \frac{\text{MSE}(f_\theta)}{n}} \right) + O \left( \sigma k\rho \sqrt{\frac{\max\{x_{\max}, 1\}\epsilon \cdot \log \left( \frac{\max\{x_{\max}, 1\} k\rho}{\epsilon\delta} \right)}{n}} \right)$$

$$+ O \left( \sigma\epsilon \sqrt{\log \left( \frac{n}{\delta} \right)} \right), \tag{80}$$

where the forth inequality results from Assumption G.6. Selecting $\epsilon = \frac{1}{k^2 \rho^2 n^2 \max\{x_{\max}, 1\}}$ and solving the second order inequality, it holds that

$$\text{MSE}(f_\theta) \leq O \left( \frac{\sigma^2 k \log \left( \frac{\max\{x_{\max}, 1\} kn\rho}{\delta} \right)}{n} \right). \tag{81}$$

$\square$

Plugging in the upper bound for MSE (Lemma G.8) to Theorem 4.1, we have the corollary below.

**Corollary G.9** (Improved version of Corollary 4.2). *For a stable solution $f = f_\theta$ of GD with step size $\eta$ where $\mathcal{L}$ is twice differentiable at $\theta$, assume Assumption G.6 holds and $f \in \mathcal{F}_\rho$ is optimized, with probability $1 - \delta$, the function $f$ satisfies*

$$\int_{-x_{\max}}^{x_{\max}} |f''(x)| g(x) dx \leq \frac{1}{\eta} - \frac{1}{2} + O \left( \sigma x_{\max} \sqrt{\frac{k \log \left( \frac{\max\{x_{\max}, 1\} kn\rho}{\delta} \right)}{n}} \right), \tag{82}$$

*where the randomness is over the noises $\{\epsilon_i\}$ and $g(x)$ is defined as (6).*

**Remark G.10.** The additional term here $\widetilde{O} \left( \sigma x_{\max} \sqrt{\frac{k}{n}} \right)$ improves over the constant additional term in Corollary 4.2 if $k < n$. In addition, if $\frac{n}{k}$ converges to infinity (*e.g.* $k = n^{1-\alpha}$ for some $\alpha > 0$ and $n$ is sufficiently large), the additional term could vanish.

# H   Proof for the Generalization Gap Bound (Theorem 4.3)

Recall that the generalization gap of function $f$ is defined as

$$\text{Gen}(f) := \left| \mathbb{E}\left[(f(x) - y)^2\right] - \frac{1}{n}\sum_{i=1}^{n}(f(x_i) - y_i)^2 \right|, \tag{83}$$

where $(x, y)$ is a new sample from the data distribution. The generalization gap measures the difference of the expected testing error and the training loss, and a small generalization gap implies that the model is not overfitting.

For a stable solution $f = f_\theta$ of GD with step size $\eta$ where $\mathcal{L}$ is twice differentiable at $\theta$, we first derive a corresponding analysis for the weighted $\text{TV}^{(1)}$ bound. Recall that the empirical loss is still defined as $\mathcal{L}(f) = \frac{1}{2n}\sum_{i=1}^{n}(f(x_i) - y_i)^2$. Then using the same calculation as (50), we have

$$
\begin{aligned}
\frac{2}{\eta} &\geq \lambda_{\max}(\nabla_\theta^2 \mathcal{L}(\theta)) \geq v^T \nabla_\theta^2 \mathcal{L}(\theta) v \\
&= \lambda_{\max}\left(\frac{1}{n}\sum_{i=1}^{n}(\nabla_\theta f_\theta(x_i))(\nabla_\theta f_\theta(x_i))^T\right) + \frac{1}{n}\sum_{i=1}^{n}(f_\theta(x_i) - y_i)v^T\nabla_\theta^2 f(x_i)v \\
&\geq \underbrace{\lambda_{\max}\left(\frac{1}{n}\sum_{i=1}^{n}(\nabla_\theta f_\theta(x_i))(\nabla_\theta f_\theta(x_i))^T\right)}_{(\star)} - \underbrace{\sqrt{\frac{1}{n}\sum_{i=1}^{n}(f_\theta(x_i) - y_i)^2} \cdot \sqrt{\frac{1}{n}\sum_{i=1}^{n}(v^T\nabla_\theta^2 f(x_i)v)^2}}_{(\#)},
\end{aligned}
\tag{84}
$$

where the last inequality results from Cauchy-Schwarz inequality.

According to Lemma G.1, the term $(\star)$ satisfies

$$(\star) = \lambda_{\max}\left(\frac{1}{n}\sum_{i=1}^{n}(\nabla_\theta f_\theta(x_i))(\nabla_\theta f_\theta(x_i))^T\right) \geq 1 + 2\int_{-x_{\max}}^{x_{\max}}|f''(x)|g(x)dx. \tag{85}$$

In addition, the term $(\#)$ satisfies (w.l.o.g, we assume $x_{\max} \geq 1$)

$$|(\#)| \leq 2x_{\max}\sqrt{\frac{1}{n}\sum_{i=1}^{n}(f_\theta(x_i) - y_i)^2}. \tag{86}$$

Under the assumption that the learned function $f = f_\theta$ satisfies $\|f\|_\infty \leq D$ and $|y_i| \leq D$ for all $i \in [n]$, it further implies that

$$|(\#)| \leq 4x_{\max}D. \tag{87}$$

Combining the results, the following $\text{TV}^{(1)}$ bound holds.

**Lemma H.1.** *Assume the data distribution satisfies that for all possible $(x, y)$ from the distribution, $|x| \leq x_{\max}$ and $|y| \leq D$, if the function $f = f_\theta$ is a stable solution of GD with step size $\eta$ such that $\mathcal{L}$ is twice differentiable at $\theta$ and $\|f\|_\infty \leq D$, then it holds that*

$$\int_{-x_{\max}}^{x_{\max}}|f''(x)|g(x)dx \leq \frac{1}{\eta} - \frac{1}{2} + 2x_{\max}D, \tag{88}$$

*where $g(x)$ is defined as (6).*

Note that we assume there exists some interval $\mathcal{I} \subset [-x_{\max}, x_{\max}]$ and a universal constant $c > 0$ such that with probability $1 - \delta/2$ (randomness over the dataset), $g(x) \geq c$ for all $x \in \mathcal{I}$ (w.l.o.g. we assume $c < 1$), which further implies that

$$\int_{\mathcal{I}}|f''(x)|dx \leq \frac{1}{c}\left(\frac{1}{\eta} - \frac{1}{2} + 2x_{\max}D\right). \tag{89}$$

We base on the high-probability event above in the following discussions. Next we bound the metric entropy of the possible output function class.

**Lemma H.2.** *Define the set* $\mathbb{T}_3 = \left\{ f : \mathcal{I} \to \mathbb{R} \,\middle|\, \|f\|_\infty \leq D, \int_\mathcal{I} |f''(x)| dx \leq \frac{1}{c}\left(\frac{1}{\eta} - \frac{1}{2} + 2x_{\max}D\right) \right\}$, *then the metric entropy with respect to* $\|\cdot\|_\infty$ *satisfies that*

$$\log N(\epsilon, \mathbb{T}_3, \|\cdot\|_\infty) \leq O\left( \sqrt{\frac{x_{\max}\left(\frac{1}{\eta} - \frac{1}{2} + 2x_{\max}D\right)}{\epsilon}} \right), \tag{90}$$

*where $O$ also absorbs the constant $c$.*

*Proof of Lemma H.2.* Define the set $\mathbb{T}_4$ as:

$$\mathbb{T}_4 = \left\{ f : [-x_{\max}, x_{\max}] \to \mathbb{R} \,\middle|\, \|f\|_\infty \leq D, \int_{-x_{\max}}^{x_{\max}} |f''(x)| dx \leq \frac{1}{c}\left(\frac{1}{\eta} - \frac{1}{2} + 2x_{\max}D\right) \right\}. \tag{91}$$

Note that the metric entropy of $\mathbb{T}_3$ is bounded by that of $\mathbb{T}_4$, therefore we directly prove the upper bound of $\log N(\epsilon, \mathbb{T}_4, \|\cdot\|_\infty)$.

Let the set $\mathbb{T}_1 = \left\{ f : [-1,1] \to \mathbb{R} \,\middle|\, \int_{-1}^1 |f''(x)| dx \leq 1, |f(x)| \leq 1 \right\}$ (as in Lemma D.4). For a fixed $\epsilon > 0$, according to Lemma D.4, there exists a $\frac{\epsilon}{\frac{1}{c}\left(\frac{1}{\eta} - \frac{1}{2} + 2x_{\max}D\right)x_{\max}}$-covering set of $\mathbb{T}_1$ with respect to $\|\cdot\|_\infty$, denoted as $\{h_i(x)\}_{i\in[N]}$, whose cardinality $N$ satisfies

$$\log N \leq C_1 \sqrt{\frac{\frac{1}{c}\left(\frac{1}{\eta} - \frac{1}{2} + 2x_{\max}D\right)x_{\max}}{\epsilon}}. \tag{92}$$

We define $g_i(x) = \frac{1}{c}\left(\frac{1}{\eta} - \frac{1}{2} + 2x_{\max}D\right)x_{\max}h_i(\frac{x}{x_{\max}})$ for all $i \in [N]$. Then $g_i$'s are all defined on $[-x_{\max}, x_{\max}]$. Obviously, we have $\{g_i(x)\}_{i\in[N]}$ also has cardinality $N$.

For any $f(x) \in \mathbb{T}_4$, we define $g(x) = \frac{1}{\frac{1}{c}\left(\frac{1}{\eta} - \frac{1}{2} + 2x_{\max}D\right)x_{\max}}f(x \cdot x_{\max})$ which is defined on $[-1,1]$. We now show that $g(x) \in \mathbb{T}_1$. First of all, for any $x \in [-x_{\max}, x_{\max}]$, we have

$$|g(x)| \leq \frac{D}{\frac{1}{c}\left(\frac{1}{\eta} - \frac{1}{2} + 2x_{\max}D\right)x_{\max}} < 1. \tag{93}$$

Meanwhile, it holds that

$$\begin{aligned}
\int_{-1}^1 |g''(x)| dx &= \int_{-1}^1 \frac{1}{\frac{1}{c}\left(\frac{1}{\eta} - \frac{1}{2} + 2x_{\max}D\right)x_{\max}} \cdot x_{\max}^2 |f''(x \cdot x_{\max})| dx \\
&\leq \frac{1}{\frac{1}{c}\left(\frac{1}{\eta} - \frac{1}{2} + 2x_{\max}D\right)} \int_{-x_{\max}}^{x_{\max}} |f''(x)| dx \leq 1.
\end{aligned} \tag{94}$$

Combining the two results, we have $g \in \mathbb{T}_1$. Therefore, there exists some $h_i$ such that $\|g - h_i\|_\infty \leq \frac{\epsilon}{\frac{1}{c}\left(\frac{1}{\eta} - \frac{1}{2} + 2x_{\max}D\right)x_{\max}}$. Since $f(x) = \frac{1}{c}\left(\frac{1}{\eta} - \frac{1}{2} + 2x_{\max}D\right)x_{\max}g(\frac{x}{x_{\max}})$, $\|g_i - f\|_\infty = \frac{1}{c}\left(\frac{1}{\eta} - \frac{1}{2} + 2x_{\max}D\right)x_{\max}\|h_i - g\|_\infty \leq \epsilon$.

In conclusion, $\{g_i\}_{i\in[N]}$ is an $\epsilon$-covering of $\mathbb{T}_4$ with respect to $\|\cdot\|_\infty$. Moreover, the cardinality of $\{g_i\}_{i\in[N]}$ is $N$, which finishes the proof. $\qquad\square$

Now we provide our main result about the generalization gap (restricted to $\mathcal{I}$). Below we define the generalization gap restricted to $\mathcal{I}$:

$$\text{Gen}_\mathcal{I}(f) := \left| \mathbb{E}_\mathcal{I}\left[ (f(x) - y)^2 \right] - \frac{1}{n_\mathcal{I}} \sum_{x_i \in \mathcal{I}} (f(x_i) - y_i)^2 \right|, \tag{95}$$

where $\mathbb{E}_\mathcal{I}$ means that $(x, y)$ is a new sample from the data distribution conditioned on $x \in \mathcal{I}$ and $n_\mathcal{I}$ is the number of data points in $\mathcal{D}$ such that $x_i \in \mathcal{I}$.

**Theorem H.3** (Restate Theorem 4.3). *Let $\mathcal{P}$ be a joint distribution of $(x, y)$ supported on $[-x_{\max}, x_{\max}] \times [-D, D]$. Assume the dataset $\mathcal{D} \sim \mathcal{P}^n$ i.i.d. For any fixed interval $\mathcal{I} \subset [-x_{\max}, x_{\max}]$ and a universal constant $c > 0$ such that with probability $1 - \delta/2$, $g(x) \geq c$ for all $x \in \mathcal{I}$, if the function $f = f_\theta$ is a stable solution of GD with step size $\eta$ such that $\mathcal{L}$ is twice differentiable at $\theta$ and $\|f\|_\infty \leq D$, with probability $1 - \delta$ (randomness over the dataset), the generalization gap restricted to $\mathcal{I}$ satisfies*

$$
\text{Gen}_{\mathcal{I}}(f) := \left| \mathbb{E}_{\mathcal{I}} \left[ (f(x) - y)^2 \right] - \frac{1}{n_{\mathcal{I}}} \sum_{x_i \in \mathcal{I}} (f(x_i) - y_i)^2 \right| \leq \widetilde{O} \left( D^{\frac{9}{5}} \left[ \frac{x_{\max} \left( \frac{1}{\eta} - \frac{1}{2} + 2 x_{\max} D \right)}{n_{\mathcal{I}}^2} \right]^{\frac{1}{5}} \right),
$$
(96)

*where $\mathbb{E}_{\mathcal{I}}$ means that $(x, y)$ is a new sample from the data distribution conditioned on $x \in \mathcal{I}$ and $n_{\mathcal{I}}$ is the number of data points in $\mathcal{D}$ such that $x_i \in \mathcal{I}$.*

*Proof of Theorem H.3.* We base on the following event that holds with probability $1 - \delta/2$:

$$
\int_{\mathcal{I}} |f''(x)| dx \leq \frac{1}{c} \left( \frac{1}{\eta} - \frac{1}{2} + 2 x_{\max} D \right).
$$
(97)

Then the output function $f \in \mathbb{T}_3$ defined in Lemma H.2.

For a fixed $\epsilon > 0$, according to Lemma H.2, there exists an $\epsilon$-covering set of $\mathbb{T}_3$ (with respect to $\|\cdot\|_\infty$) whose cardinality $N$ satisfies that

$$
\log N \leq O \left( \sqrt{\frac{x_{\max} \left( \frac{1}{\eta} - \frac{1}{2} + 2 x_{\max} D \right)}{\epsilon}} \right).
$$
(98)

For a fixed function $\bar{f}$ in the covering set, since the data set $\{(x_i, y_i)\}_{x_i \in \mathcal{I}}$ is still i.i.d. from the data distribution conditioned on $x \in \mathcal{I}$, Hoeffding's inequality (Lemma K.3) implies that with probability $1 - \delta$, it holds that

$$
\left| \mathbb{E}_{\mathcal{I}} \left[ (\bar{f}(x) - y)^2 \right] - \frac{1}{n_{\mathcal{I}}} \sum_{x_i \in \mathcal{I}} (\bar{f}(x_i) - y_i)^2 \right| \leq 4 D^2 \cdot \sqrt{\frac{\log(2/\delta)}{n_{\mathcal{I}}}}.
$$
(99)

Together with a union bound over the covering set, we have with probability $1 - \delta/2$, for all $\bar{f}$ in the covering set,

$$
\left| \mathbb{E}_{\mathcal{I}} \left[ (\bar{f}(x) - y)^2 \right] - \frac{1}{n_{\mathcal{I}}} \sum_{x_i \in \mathcal{I}} (\bar{f}(x_i) - y_i)^2 \right| \leq 4 D^2 \cdot \sqrt{\frac{\log(4N/\delta)}{n_{\mathcal{I}}}}
$$

$$
\leq O \left( D^2 \cdot \frac{\left[ x_{\max} \left( \frac{1}{\eta} - \frac{1}{2} + 2 x_{\max} D \right) \right]^{\frac{1}{4}} \log(1/\delta)^{\frac{1}{2}}}{n_{\mathcal{I}}^{\frac{1}{2}} \epsilon^{\frac{1}{4}}} \right).
$$
(100)

Under such high probability event, for any $f \in \mathbb{T}_3$, let $\bar{f}$ be a function in the covering set such that $\|f - \bar{f}\|_\infty \leq \epsilon$. Then it holds that

$$
\left| \mathbb{E}_{\mathcal{I}} \left[ (f(x) - y)^2 \right] - \frac{1}{n_{\mathcal{I}}} \sum_{x_i \in \mathcal{I}} (f(x_i) - y_i)^2 \right|
$$

$$
\leq \left| \mathbb{E}_{\mathcal{I}} \left[ (\bar{f}(x) - y)^2 \right] - \frac{1}{n_{\mathcal{I}}} \sum_{x_i \in \mathcal{I}} (\bar{f}(x_i) - y_i)^2 \right| + O(D\epsilon)
$$

$$
\leq O(D\epsilon) + O \left( D^2 \cdot \frac{\left[ x_{\max} \left( \frac{1}{\eta} - \frac{1}{2} + 2 x_{\max} D \right) \right]^{\frac{1}{4}} \log(1/\delta)^{\frac{1}{2}}}{n_{\mathcal{I}}^{\frac{1}{2}} \epsilon^{\frac{1}{4}}} \right)
$$
(101)

$$
\leq O \left( D^{\frac{9}{5}} \left[ \frac{x_{\max} \left( \frac{1}{\eta} - \frac{1}{2} + 2 x_{\max} D \right) \log(1/\delta)^2}{n_{\mathcal{I}}^2} \right]^{\frac{1}{5}} \right),
$$

where the last inequality results from selecting the $\epsilon$ that minimizes the objective. $\square$

## H.1 Choice of the Interval Under Uniform Distribution

In this part, we discuss the choice of the interval $\mathcal{I}$ under the case that the marginal distribution of $x$ is the uniform distribution on $[-x_{\max}, x_{\max}]$. For simplicity, we assume that $x_{\max} = 1$.

**Lemma H.4.** *Assume that $x \sim \mathrm{Unif}([-1, 1])$, then we can choose $\mathcal{I}$ to be $[-\frac{2}{3}, \frac{2}{3}]$. In this way, with probability $1 - 24e^{-\frac{n}{96}}$, for any $x \in \mathcal{I}$, it holds that*

$$g(x) \geq \frac{1}{4320}. \tag{102}$$

*As a result, when $n \geq 96 \log\left(\frac{48}{\delta}\right)$, we can choose $\mathcal{I} = [-\frac{2}{3}, \frac{2}{3}]$ and $c = \frac{1}{4320}$.*

*Proof of Lemma H.4.* Let the intervals $A_i$ be defined as below: for all $i \in [12]$,

$$A_i = \left[\frac{i-7}{6}, \frac{i-6}{6}\right]. \tag{103}$$

For a fixed $n$, let $P_i$ denote the number of data points in $A_i$, which follows Binomial distribution with $p = \frac{1}{12}$. Then for a fixed $i \in [12]$, according to Multiplicative Chernoff bound (Lemma K.4), with probability $1 - 2e^{-\frac{n}{96}}$, it holds that

$$\frac{1}{2} \cdot \frac{n}{12} \leq P_i \leq 2 \cdot \frac{n}{12}. \tag{104}$$

Then according to a union bound, with probability $1 - 24e^{-\frac{n}{96}}$, the above inequality holds for all $i \in [12]$. Under the case above, we prove that $g(x) = \min\{g^-(x), g^+(x)\} \geq \frac{1}{4320}$ for all $x \in \mathcal{I}$.

Recall that $g^-(x) = \mathbb{P}^2(X < x)\mathbb{E}[x - X | X < x]\sqrt{1 + (\mathbb{E}[X | X < x])^2}$ where $X$ is drawn from the empirical distribution of the data (a sample chosen uniformly from $\{x_j\}$). Then for any $x \geq -\frac{2}{3}$,

$$\mathbb{P}^2(X < x) \geq \left(\frac{P_1 + P_2}{n}\right)^2 \geq \frac{1}{144},$$

$$\mathbb{E}[x - X | X < x] \geq \frac{\frac{n}{24} \cdot \frac{1}{6}}{\frac{n}{24} + \frac{n}{6}} = \frac{1}{30}, \tag{105}$$

$$\sqrt{1 + (\mathbb{E}[X | X < x])^2} \geq 1.$$

Combining the inequalities, we have $g^-(x) \geq \frac{1}{4320}$ for all $x \in \mathcal{I}$. The result for $g^+(x)$ can be proven similarly, which implies that with probability $1 - 24e^{-\frac{n}{96}}$, $g(x) \geq \frac{1}{4320}$ for all $x \in \mathcal{I}$.

The implication can be proven by direct calculation. $\square$

**Remark H.5.** We only consider the case where the data is sampled from uniform distribution, while we remark that for various distributions that are not heavy-tailed (*e.g.* Gaussian distribution, Laplace distribution), a similar result can be derived. Some empirical illustrations are shown in Appendix H.2 below.

## H.2 More Illustrations of the Choice of the Interval

In this part, we consider the choice of $\mathcal{I}$ and $c$ under different data distributions. More specifically, we consider the following four distributions of $x$.

**Uniform distribution**: $x \sim \mathrm{Unif}([-1, 1])$.

**Normal distribution**: $x \sim \mathcal{N}(0, 1)$.

**Laplace distribution**: $x \sim \mathrm{Laplace}(0, 1)$.

**Gaussian mixture distribution**: $x \sim \begin{cases} \mathcal{N}(-0.5, 0.25) & \text{with probability } \frac{1}{2}, \\ \mathcal{N}(0.5, 0.25) & \text{with probability } \frac{1}{2}. \end{cases}$

For each distribution, we sample $n = 1000$ data points from the distribution (conditional on $x \in [-1, 1]$) and construct the $g(x)$ function. Then we choose the interval $\mathcal{I}$ and the corresponding lower bound $c$ of $g(x)$ over $\mathcal{I}$. From Figure 9, we find that for all of the four distributions, with a constant $c \geq 0.002$, the interval $\mathcal{I}$ can be chosen to incorporate a large portion of the data ($n_{\mathcal{I}} \geq 0.65n$).

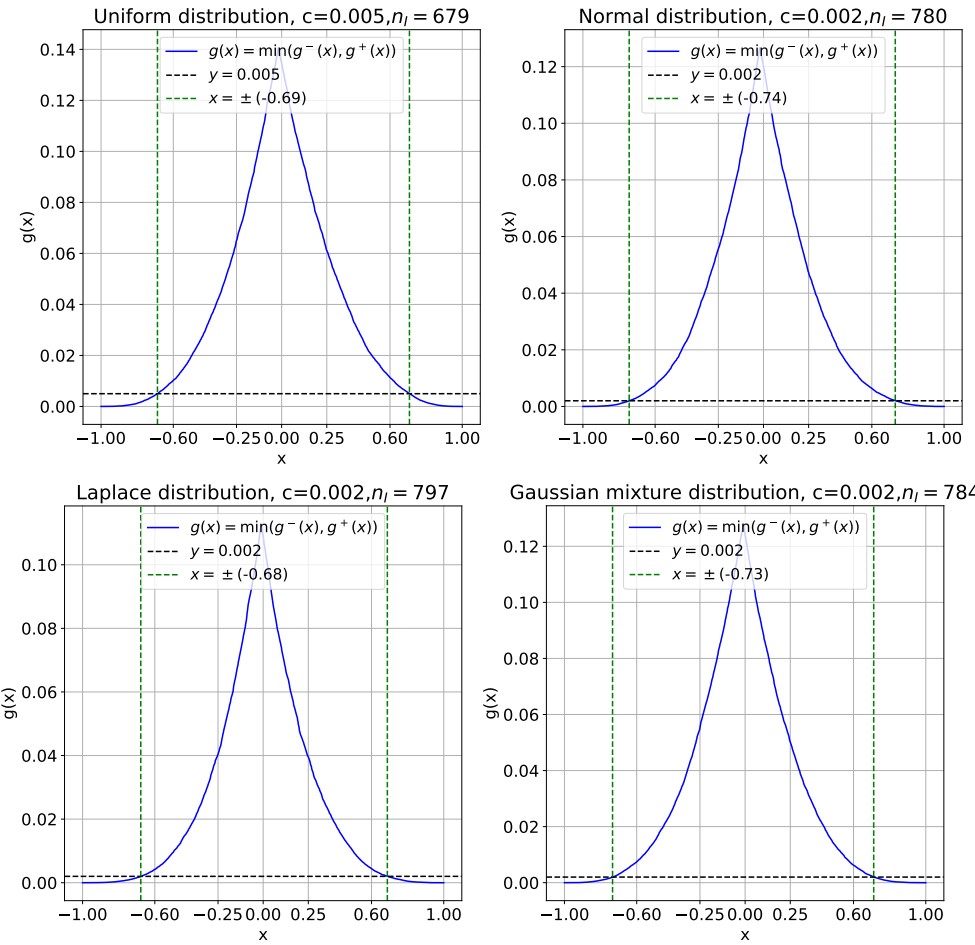

Figure 9: Illustration of the choice of interval $\mathcal{I}$ and the corresponding lower bound $c$ for $g(x)$.

## I  Proof for the Refined MSE Bound (Theorem 4.4)

In this part, we base on the same conditions in Corollary 4.2, which is the weighted $\mathrm{TV}^{(1)}$ upper bound. For an output stable solution $f$ satisfying the conclusion of Corollary 4.2, we have $\int_{-x_{\max}}^{x_{\max}} |f''(x)| g(x) dx \leq \frac{1}{\eta} - \frac{1}{2} + \widetilde{O}(\sigma x_{\max})$ and we denote the right-hand side by $S$. In addition, according to the assumption that $g(x) \geq c$ for any $x \in \mathcal{I}$, we further have $\int_{\mathcal{I}} |f''(x)| dx \leq \frac{S}{c}$. Now we bound the complexity of the possible output function class.

According to the definition of two-layer ReLU network and our assumption that $\|\theta\|_\infty \leq \rho$, it holds that:

$$|f(0)| = \left| \sum_{i=1}^{k} w_i^{(2)} \phi\left(b_i^{(1)}\right) + b^{(2)} \right| \leq k\rho^2 + \rho. \tag{106}$$

$$|f'(0)| \leq \sum_{i=1}^{k} \left| w_i^{(2)} w_i^{(1)} \right| \leq k\rho^2. \tag{107}$$

Define the set $\mathbb{T} = \left\{ f : \mathcal{I} \to \mathbb{R} \mid |f(0)| \leq k\rho^2 + \rho, \ |f'(0)| \leq k\rho^2, \ \int_{\mathcal{I}} |f''(x)| dx \leq \frac{S}{c} \right\}$. According to the inequalities above, the possible output function (if restricted to $\mathcal{I}$) belongs to $\mathbb{T}$. We begin with an analysis of the metric entropy of the intermediate function set $\mathbb{T}_2$.

**Lemma I.1.** *Assume the set* $\mathbb{T}_2 = \left\{ f : [-x_{\max}, x_{\max}] \to \mathbb{R} \mid f(0) = f'(0) = 0, \ \int_{-x_{\max}}^{x_{\max}} |f''(x)| dx \leq C_2 \right\}$ *for some constant* $C_2 > 0$, *and the metric is* $\ell_\infty$ *distance* $\|\cdot\|_\infty$, *then there exists a universal constant*

$C_1 > 0$ such that for any $\epsilon > 0$, the metric entropy of $(\mathbb{T}_2, \|\cdot\|_\infty)$ satisfies

$$\log N(\epsilon, \mathbb{T}_2, \|\cdot\|_\infty) \leq C_1 \sqrt{\frac{C_2 x_{\max}}{\epsilon}}, \tag{108}$$

where $C_1$ can be chosen as the same $C_1$ in Lemma D.4.

*Proof of Lemma I.1.* Let the set $\mathbb{T}_1 = \left\{ f : [-1,1] \to \mathbb{R} \;\middle|\; \int_{-1}^{1} |f''(x)| dx \leq 1, \; |f(x)| \leq 1 \right\}$ (as in Lemma D.4). For a fixed $\epsilon > 0$, according to Lemma D.4, there exists a $\frac{\epsilon}{C_2 x_{\max}}$-covering set of $\mathbb{T}_1$ with respect to $\|\cdot\|_\infty$, denoted as $\{h_i(x)\}_{i \in [N]}$, whose cardinality $N$ satisfies

$$\log N \leq C_1 \sqrt{\frac{C_2 x_{\max}}{\epsilon}}. \tag{109}$$

We define $g_i(x) = C_2 x_{\max} h_i(\frac{x}{x_{\max}})$ for all $i \in [N]$. Then $g_i$'s are all defined on $[-x_{\max}, x_{\max}]$. Obviously, we have $\{g_i(x)\}_{i \in [N]}$ also has cardinality $N$.

For any $f(x) \in \mathbb{T}_2$, we define $g(x) = \frac{1}{C_2 x_{\max}} f(x \cdot x_{\max})$ which is defined on $[-1,1]$. We now show that $g(x) \in \mathbb{T}_1$. First of all, for any $x \in [-x_{\max}, x_{\max}]$, we have $|f'(x)| \leq \int_{-x_{\max}}^{x_{\max}} |f''(x)| dx \leq C_2$. Therefore, for any $x \in [-x_{\max}, x_{\max}]$, $|f(x)| \leq C_2 x_{\max}$, which implies that $|g(x)| \leq 1$ for any $x \in [-1,1]$. Meanwhile, it holds that

$$\begin{aligned}
\int_{-1}^{1} |g''(x)| dx &= \int_{-1}^{1} \frac{1}{C_2 x_{\max}} \cdot x_{\max}^2 |f''(x \cdot x_{\max})| dx \\
&\leq \frac{1}{C_2} \int_{-x_{\max}}^{x_{\max}} |f''(x)| dx \leq 1.
\end{aligned} \tag{110}$$

Combining the two results, we have $g \in \mathbb{T}_1$. Therefore, there exists some $h_i$ such that $\|g - h_i\|_\infty \leq \frac{\epsilon}{C_2 x_{\max}}$. Since $f(x) = C_2 x_{\max} g(\frac{x}{x_{\max}})$, $\|g_i - f\|_\infty = C_2 x_{\max} \|h_i - g\|_\infty \leq \epsilon$.

In conclusion, $\{g_i\}_{i \in [N]}$ is an $\epsilon$-covering of $\mathbb{T}_2$ with respect to $\|\cdot\|_\infty$. Moreover, the cardinality of $\{g_i\}_{i \in [N]}$ is $N$, which finishes the proof. $\qquad \square$

With Lemma I.1, we are ready to bound the metric entropy of $\mathbb{T}$.

**Lemma I.2.** *Assume the metric is $\ell_\infty$ distance $\|\cdot\|_\infty$, then the metric entropy of $(\mathbb{T}, \|\cdot\|_\infty)$ satisfies*

$$\log N(\epsilon, \mathbb{T}, \|\cdot\|_\infty) \leq O\left( \sqrt{\frac{x_{\max} S}{\epsilon}} \right), \tag{111}$$

where $S$ is the right-hand side of Corollary 4.2 and $O$ also absorbs the constant $c$.

*Proof of Lemma I.2.* For any function $f \in \mathbb{T}$, it can be written as below:

$$f(x) = f(0) + f'(0)x + g(x), \tag{112}$$

where $g(x) = f(x) - f(0) - f'(0)x$ satisfies that $g(0) = g'(0) = 0$ and $g''(x) = f''(x)$. Therefore, to cover $\mathbb{T}$ to $\epsilon$ accuracy, it suffices to cover the three parts to $\frac{\epsilon}{3}$ accuracy with respect to $\|\cdot\|_\infty$, respectively.

For $f(0)$, since $|f(0)| \leq k\rho^2 + \rho$, the covering number is bounded by

$$N_1 \leq \frac{6(k\rho^2 + \rho)}{\epsilon} \leq \frac{8k\rho^2}{\epsilon}. \tag{113}$$

For $f'(0)x$, since $|f'(0)| \leq k\rho^2$, the covering number is bounded by

$$N_2 \leq \frac{6k\rho^2 x_{\max}}{\epsilon}. \tag{114}$$

Finally, for $g(x)$, since $g(x) \in \mathbb{T}_2$ with $C_2 = \frac{S}{c}$ ($g$ is extended linearly beyond the interval $\mathcal{I}$), the covering number is bounded according to Lemma I.1 above.

$$\log N_3 \leq C_1 \sqrt{\frac{3x_{\max}S}{c\epsilon}}. \tag{115}$$

Combining the three parts, the metric entropy is bounded by

$$\log N(\epsilon, \mathbb{T}, \|\cdot\|_\infty) \leq \log N_1 + \log N_2 + \log N_3 \leq O\left(\sqrt{\frac{x_{\max}S}{\epsilon}}\right), \tag{116}$$

where $O$ also absorbs $c$, which is the constant lower bound of $g(x)$. $\qquad\square$

According to the metric entropy above, we are ready to provide a refined (high probability) bound for mean squared error (restricted to $\mathcal{I}$). Note that we assume that the ground-truth function $f_0 \in \mathbb{T}$, which is necessary for the mean squared error to vanish.

**Lemma I.3.** *Under the same conditions in Corollary 4.2, for any interval $\mathcal{I} \subset [-x_{\max}, x_{\max}]$ and a universal constant $c > 0$ such that $g(x) \geq c$ for all $x \in \mathcal{I}$ and $f$ is optimized over $\mathcal{I}$, i.e. $\sum_{x_i \in \mathcal{I}}(f(x_i) - y_i)^2 \leq \sum_{x_i \in \mathcal{I}}(f_0(x_i) - y_i)^2$, if the output stable solution $\theta$ satisfies $\|\theta\|_\infty \leq \rho$ and the ground truth $f_0 \in \mathbb{T}$, then with probability $1 - \delta$ (over the random noises $\{\epsilon_i\}$), the function $f = f_\theta$ satisfies*

$$\mathrm{MSE}_\mathcal{I}(f) = \frac{1}{n_\mathcal{I}}\sum_{x_i \in \mathcal{I}}(f(x_i) - f_0(x_i))^2 \leq O\left(\left(\frac{\sigma^2}{n_\mathcal{I}}\right)^{\frac{4}{5}}(x_{\max}S)^{\frac{2}{5}}\log\left(\frac{1}{\delta}\right)\right), \tag{117}$$

*where $n_\mathcal{I}$ is the number of data points in $\mathcal{D}$ such that $x_i \in \mathcal{I}$.*

*Proof of Lemma I.3.* According to the assumption that $f$ is optimized over $\mathcal{I}$, we have

$$\sum_{x_i \in \mathcal{I}}(f(x_i) - y_i)^2 \leq \sum_{x_i \in \mathcal{I}}(f_0(x_i) - y_i)^2. \tag{118}$$

Similar to the calculation in Lemma G.8, it holds that

$$\frac{1}{2}\mathrm{MSE}_\mathcal{I}(f) = \frac{1}{2n_\mathcal{I}}\sum_{x_i \in \mathcal{I}}(f(x_i) - f_0(x_i))^2 \leq \frac{1}{n_\mathcal{I}}\sum_{x_i \in \mathcal{I}}\epsilon_i\,(f(x_i) - f_0(x_i)). \tag{119}$$

It is obvious that the function class $\mathbb{T}$ is convex, together with the assumption that $f_0 \in \mathbb{T}$, we have the function set $\mathbb{T}^\star := \mathbb{T} - \{f_0\}$ is star-shaped (details in Section 13 of Wainwright [2019]).

Note that the metric entropy of $\mathbb{T}$ satisfies that $\log N(\epsilon, \mathbb{T}, \|\cdot\|_\infty) \leq O\left(\sqrt{\frac{x_{\max}S}{\epsilon}}\right)$. According to Corollary 13.7 of Wainwright [2019], the critical radius $r$ satisfies that

$$r^2 \leq O\left(\left(\frac{\sigma^2}{n_\mathcal{I}}\right)^{\frac{4}{5}}(x_{\max}S)^{\frac{2}{5}}\right). \tag{120}$$

Finally, according to Theorem 13.5 of Wainwright [2019], we have with probability $1 - \delta$,

$$\mathrm{MSE}_\mathcal{I}(f) \leq O\left(\left(\frac{\sigma^2}{n_\mathcal{I}}\right)^{\frac{4}{5}}(x_{\max}S)^{\frac{2}{5}}\log\left(\frac{1}{\delta}\right)\right), \tag{121}$$

which finishes the proof. $\qquad\square$

Finally, Theorem 4.4 is derived by plugging in the definition of $S$.

**Theorem I.4** (Restate Theorem 4.4). *Under the same conditions in Corollary 4.2, for any interval $\mathcal{I} \subset [-x_{\max}, x_{\max}]$ and a universal constant $c > 0$ such that $g(x) \geq c$ for all $x \in \mathcal{I}$ and $f$ is optimized over $\mathcal{I}$, i.e. $\sum_{x_i \in \mathcal{I}} (f(x_i) - y_i)^2 \leq \sum_{x_i \in \mathcal{I}} (f_0(x_i) - y_i)^2$, if the output stable solution $\theta$ satisfies $\|\theta\|_\infty \leq \rho$ (for some constant $\rho > 0$) and the ground truth $f_0 \in \mathrm{BV}^{(1)}(k\rho^2, \frac{1}{c}\widetilde{O}(\frac{1}{\eta} + \sigma x_{\max}))$, then with probability $1 - \delta$ (over the random noises $\{\epsilon_i\}$), the function $f = f_\theta$ satisfies*

$$\mathrm{MSE}_{\mathcal{I}}(f) = \frac{1}{n_{\mathcal{I}}} \sum_{x_i \in \mathcal{I}} (f(x_i) - f_0(x_i))^2 \leq \widetilde{O}\left( \left(\frac{\sigma^2}{n_{\mathcal{I}}}\right)^{\frac{4}{5}} \left(\frac{x_{\max}}{\eta} + \sigma x_{\max}^2\right)^{\frac{2}{5}} \right), \tag{122}$$

*where $n_{\mathcal{I}}$ is the number of data points in $\mathcal{D}$ such that $x_i \in \mathcal{I}$.*

*Proof of Theorem I.4.* Note that $\mathrm{BV}^{(1)}(k\rho^2, \frac{1}{c}\widetilde{O}(\frac{1}{\eta} + \sigma x_{\max}))$ is a subset of $\mathbb{T}$. Then the proof directly results from Lemma I.3 and $S = \widetilde{O}\left(\frac{1}{\eta} + \sigma x_{\max}\right)$. $\qquad \square$

## I.1 The Improved Results for the Under-parameterized Case

We assume that $n/k$ is large enough such that the additional term in the $\mathrm{TV}^{(1)}$ bound vanishes.

**Assumption I.5.** We assume that $\frac{n}{k}$ is large enough such that the last term in (82) $\widetilde{O}(\sigma x_{\max}\sqrt{k/n}) \leq \frac{1}{2}$, which further implies that $\int_{-x_{\max}}^{x_{\max}} |f''(x)|g(x)dx \leq \frac{1}{\eta}$, where $g(x)$ is defined as (6).

Assumption I.5 requires that $\frac{n}{k}$ is larger than some constant, which naturally holds if $k = n^{1-\alpha}$ for some $\alpha > 0$ and $n$ is sufficiently large. Under such assumption, we improve the MSE upper bound.

**Theorem I.6.** *Under the same conditions in Corollary G.9, assume that Assumption I.5 holds. For any interval $\mathcal{I} \subset [-x_{\max}, x_{\max}]$ and a universal constant $c > 0$ such that $g(x) \geq c$ for all $x \in \mathcal{I}$ and $f$ is optimized over $\mathcal{I}$, i.e. $\sum_{x_i \in \mathcal{I}} (f(x_i) - y_i)^2 \leq \sum_{x_i \in \mathcal{I}} (f_0(x_i) - y_i)^2$, if the output stable solution $\theta$ satisfies $\|\theta\|_\infty \leq \rho$ (for some constant $\rho > 0$) and the ground truth $f_0 \in \mathrm{BV}^{(1)}(k\rho^2, \frac{1}{c\eta})$, then with probability $1 - \delta$ (over the random noises $\{\epsilon_i\}$), the function $f = f_\theta$ satisfies*

$$\mathrm{MSE}_{\mathcal{I}}(f) = \frac{1}{n_{\mathcal{I}}} \sum_{x_i \in \mathcal{I}} (f(x_i) - f_0(x_i))^2 \leq \widetilde{O}\left( \left(\frac{\sigma^2}{n_{\mathcal{I}}}\right)^{\frac{4}{5}} \left(\frac{x_{\max}}{\eta}\right)^{\frac{2}{5}} \right), \tag{123}$$

*where $n_{\mathcal{I}}$ is the number of data points in $\mathcal{D}$ such that $x_i \in \mathcal{I}$.*

*Proof of Theorem I.6.* The proof is identical to Theorem I.4, with $S$ replaced by $\frac{1}{\eta}$. $\qquad \square$

**Remark I.7.** Compared to Theorem 4.4, Theorem I.6 is better on the dependence of $\eta$ by removing the additional term $\sigma x_{\max}^2$. Such improvement results from the improved $\mathrm{TV}^{(1)}$ bound (Corollary G.9) and the fact that $n/k$ is sufficiently large.

## J Twice-Differentiable Interpolating Solution with Noisy Labels must be "Sharp"

Recall that in the counter-example, we fix $x_i = \frac{2x_{\max}i}{n-1} - \frac{(n+1)x_{\max}}{n-1}$ for $i \in [n]$ and $f_0(x) = 0$ for any $x$, which implies that $y_i$'s are independent random variables from $\mathcal{N}(0, \sigma^2)$.

**Proposition J.1.** *In the example above, assume $f = f_\theta$ is an interpolating solution where $\mathcal{L}$ is twice differentiable at $\theta$, then with probability $1 - \delta$, we have*

$$\lambda_{\max}(\nabla_\theta^2 \mathcal{L}(\theta)) = \Omega\left( \sigma n \left[ n - 24 \log\left(\frac{1}{\delta}\right) \right] \right). \tag{124}$$

*Proof of Proposition J.1.* According to Theorem 3.1, with probability $1 - \delta$, we have

$$\int_{-x_{\max}}^{x_{\max}} |f_\theta''(x)|g(x)dx = \Omega\left( \sigma n \left[ n - 24 \log\left(\frac{1}{\delta}\right) \right] \right), \tag{125}$$

where $g(x)$ is defined in (6). Meanwhile, note that $f_\theta$ is an interpolating solution, and therefore

$$
\begin{aligned}
\nabla_\theta^2 \mathcal{L}(\theta) =& \frac{1}{n} \sum_{i=1}^n (\nabla_\theta f_\theta(x_i))(\nabla_\theta f_\theta(x_i))^T + \frac{1}{n} \sum_{i=1}^n (f_\theta(x_i) - y_i) \nabla_\theta^2 f(x_i) \\
=& \frac{1}{n} \sum_{i=1}^n (\nabla_\theta f_\theta(x_i))(\nabla_\theta f_\theta(x_i))^T.
\end{aligned}
\tag{126}
$$

Finally, combining the results and applying Lemma G.1, it holds that

$$
\begin{aligned}
\lambda_{\max}(\nabla_\theta^2 \mathcal{L}(\theta)) &= \lambda_{\max} \left( \frac{1}{n} \sum_{i=1}^n (\nabla_\theta f_\theta(x_i))(\nabla_\theta f_\theta(x_i))^T \right) \\
&\geq 1 + 2 \int_{-x_{\max}}^{x_{\max}} |f_\theta''(x)| g(x) dx \geq \Omega \left( \sigma n \left[ n - 24 \log \left( \frac{1}{\delta} \right) \right] \right),
\end{aligned}
\tag{127}
$$

which finishes the proof. $\qquad \square$

## K   Technical Lemmas

**Lemma K.1** (Lemma F.4 in Dann et al. [2017])**.** *Let $F_i$ for $i = 1, \cdots$ be a filtration and $X_1, \cdots, X_n$ be a sequence of Bernoulli random variables with $\mathbb{P}(X_i = 1 | F_{i-1}) = P_i$ with $P_i$ being $F_{i-1}$-measurable and $X_i$ being $F_i$ measurable. It holds that*

$$
\mathbb{P} \left[ \exists n : \sum_{t=1}^n X_t < \sum_{t=1}^n P_t / 2 - W \right] \leq e^{-W}.
$$

**Lemma K.2** (Covering Number of Euclidean Ball [Wainwright, 2019])**.** *For any $\epsilon > 0$, the $\epsilon$-covering number of the Euclidean ball in $\mathbb{R}^d$ with radius $R > 0$ is upper bounded by $(1 + \frac{2R}{\epsilon})^d$.*

**Lemma K.3** (Hoeffding's inequality [Sridharan, 2002])**.** *Suppose $X_1, X_2, \cdots, X_n$ are a sequence of independent, identically distributed (i.i.d.) random variables with mean 0. Let $\bar{X} = \frac{1}{n} \sum_{i=1}^n X_i$. Suppose that $X_i \in [-b, b]$ with probability 1, then with probability $1 - \delta$,*

$$
|\bar{X}| \leq b \cdot \sqrt{\frac{2 \log(2/\delta)}{n}}.
\tag{128}
$$

**Lemma K.4** (Multiplicative Chernoff bound [Chernoff, 1952])**.** *Let $X$ be a Binomial random variable with parameters $p, n$. Then for any $\delta \in [0, 1]$, it holds that:*

$$
\mathbb{P}[X > (1 + \delta)pn] < e^{-\frac{\delta^2 pn}{3}},
\tag{129}
$$

$$
\mathbb{P}[X < (1 - \delta)pn] < e^{-\frac{\delta^2 pn}{2}}.
\tag{130}
$$

The lemmas above are also applied in Qiao et al. [2022], Qiao and Wang [2023a,c], Qiao et al. [2023], Qiao and Wang [2023b, 2024], Zhao et al. [2022], Xu et al. [2023].

