# OpenReview forum: "Stable Minima Cannot Overfit in Univariate ReLU Networks: Generalization by Large Step Sizes"
_NeurIPS.cc/2024/Conference — NeurIPS 2024 spotlight_

### Official Review · Reviewer_3THJ · 2024-06-28

**Soundness:** 4
**Presentation:** 4
**Contribution:** 4
**Rating:** 8
**Confidence:** 4

**Summary:**

This paper studies the implicit regularization of large learning rates in gradient descent. The setting is univariate linear regression with two-layer ReLU neural networks. The authors show that, if GD converges to a local minimum, then the function implemented by the neural network at this local minimum has a bounded first-order total variation. This regularity avoids overfitting, as quantified by a generalization bound. Numerical experiments validate the results.

**Strengths:**

Understanding the implicit regularization of optimization algorithms for deep learning is a key topic. This paper studies a fairly realistic setting, which does not necessitate interpolation. To my knowledge, the results are novel and go beyond what was already known on the impact of large learning rates (edge of stability, minima stability). Most previous works focus on parameter space and not function space. The interpretation in terms of function space proposed in the present paper is interesting. The paper is very clearly-written. While I have not read the proofs, the mathematical presentation of the results in the main text is very precise.

**Weaknesses:**

The paper considers a univariate case, some comments on the extension to the multivariate case are given at the end of Section 1.1.

**Questions:**

Do the authors have any insight on the role of overparameterization in the setting they consider? It does not seem to have a large influence on the generalization bounds, but perhaps it would help the optimization (e.g., via a PL inequality)?

Minor remarks:
- lines 132: the connection between stepsize and L1/L2 regularization has been thoroughly analysed for diagonal linear networks in [2].
- lines 147-153: a relevant theoretical paper on the Edge of Stability is [1].

[1] Damin, Nichani, Lee, Self-Stabilization: The Implicit Bias of Gradient Descent at the Edge of Stability, ICLR 2023.

[2] Even, Pesme, Gunasekar, Flammarion. (S)GD over Diagonal Linear Networks: Implicit Regularisation, Large Stepsizes and Edge of Stability. NeurIPS 2023.

**Limitations:**

Limitations are properly addressed.

---

> ### Author Rebuttal · Authors · 2024-08-06
>
> We appreciate your high quality review and the positive score. Below we will reply to your comments.
>
> **The paper considers a univariate case, some comments on the extension to the multivariate case are given at the end of Section 1.1.**
>
> Since this is the first paper to consider minima stability without the assumption of interpolation, we start with the univariate case. The generalization to multivariate functions includes deriving a similar weighted TV(1) bound (with $f^{\prime\prime}(x)$ replaced by the Laplacian $\Delta(f)$). Based on the analysis for the multivariate case with interpolation [Nacson et al 2022], the additional term about approximation error needs to be handled as in this submission. With a TV(1) bound, the learned function belongs to a Radon TV class, and the remaining analysis is based on the metric entropy of such a function class. We believe this to be an interesting and promising future direction.
>
> **Do the authors have any insight on the role of overparameterization in the setting they consider? It does not seem to have a large influence on the generalization bounds, but perhaps it would help the optimization (e.g., via a PL inequality)?**
>
> This is a very good question.
>
> Overparameterization ensures that the neural network is able to ''approximate'' the underlying function $f_0$, i.e., there exist neural networks such that the training loss can be smaller than $\sigma^2$. It is also generally believed that overparameterization makes optimization easy.  In our numerical experiments, we found that underparameterized NNs (with just a few knots) are more likely to get stuck at solutions with knots in awkward locations. On the other hand, when the NNs are overparameterized, even at initialization, there are already candidate basis functions with "knots" near every input data point, thus making it much easier to find good solutions that are ''optimized''. This is a hypothesis that our experiments (in Figure 4,5) supports.
>
> Our generalization bounds work for both overparameterized and underparameterized NNs. The theoretical insight is that, when trained with GD, overparameterization does not cause overfitting, which offers new theoretical justifiction to how GD-trained overparameterized NN works even though there are more than enough parameters to overfit the noise.
>
> As for the PL condition:  it will be very interesting if one can obtain PL condition (or any approximate variant of that). Existing work on the optimization of overparameterized NN focuses on the ''interpolation'' regime which does not apply to our problem.
>
>
> **Regarding the references.**
>
> Thanks for pointing us to these references. We will add discussions about these papers in the next version.
>
> Thanks again for the high-quality review. We hope our response could address your main concerns and we are happy to answer any further questions.

---

> > ### Comment · Reviewer_3THJ · 2024-08-07
> > **Thank you for the rebuttal.**
> >
> > I thank the authors for their rebuttal and keep my score.

---

> > > ### Author Response · Authors · 2024-08-11
> > > **Thank you!**
> > >
> > > Thanks for acknowledging our response and for your continued support!

---

### Official Review · Reviewer_4Yyh · 2024-07-13

**Soundness:** 3
**Presentation:** 4
**Contribution:** 3
**Rating:** 8
**Confidence:** 3

**Summary:**

This paper studies the generalization properties of two-layer ReLU neural networks in a univariate nonparametric regression problem with noisy labels. It proposes a new theory for local minima to which gradient descent (GD) with a fixed learning rate $\eta$ stably converges. The paper shows that GD with a constant learning rate can only find stable local minima whose weighted TV(1) is bounded by $1/\eta-1/2+\tilde{O}(\sigma+\sqrt{\mathrm{MSE}})$. With this property of local minimas, they prove the generalization bound for univariate nonparametric regression. The theoretical results are validated by extensive simulations demonstrating that large learning rate training induces sparse linear spline fits.

**Strengths:**

This paper gives an end-to-end analysis of the generalization of two layer ReLU networks on learning nonparametric univariate functions. The completeness of the work is significant. The theoretical analysis is solid and the proof is well-organized. The authors also present extensive experiments to support their theoretical findings. In general, it is a good paper.

**Weaknesses:**

1. The analysis in this paper has little focus on optimization. There is no rigorous theoretical evidence that GD will find the solutions that satisfies the assumptions (though I know this is an open problem in the literature).
2. This paper is limited to univariate functions, which is not important in practice. The authors claims that the technique can be generated to multivariate functions but they did not show that.

**Questions:**

How large is the family of target functions satisfying the BV(1) condition?

**Limitations:**

The authors have addressed some limitations of their work, acknowledging that their analysis is only for full batch gradient descent and univariate nonparamatric regression.

---

> ### Author Rebuttal · Authors · 2024-08-06
>
> We appreciate your high quality review and the positive score. Below we will reply to your comments.
>
> **The analysis in this paper has little focus on optimization. There is no rigorous theoretical evidence that GD will find the solutions that satisfies the assumptions (though I know this is an open problem in the literature).**
>
> We agree with your point. Line 94-105 in the paper was written with the hope of avoiding the confusion w.r.t. the optimization.
>
> Our results focus on the generalization of the neural nets GD can stably find. We do not have new computational guarantees on GD convergence. Convergence of GD for training a neural network in non-interpolating regime (to an ''optimized'' solution) is an open problem that would probably require additional assumptions.
>
> To the defense of our paper, almost all statistical learning theory is about ERM or regularized ERM which does not deal with optimization at all. The interesting discovery from our paper is that we not only do not require ERM, we showed that ERMs are very bad.
>
> While we cannot show that GD finds good solutions, we proved that (1) GD cannot converge to interpolating solutions (unless the learning rate is tiny) (2) all solutions GD can converge to cannot overfit.  These are, in our opinion, strong and new theoretical results about ''optimization'' even though they are not about computation but rather about ''optimization-induced'' solutions.
>
> Moreover, we find that our bounds hold for a wider range of cases in the sense that the convergence of GD is not necessary. The weighted TV(1) bound of $f_\theta$ is valid as well as the corresponding $\theta$ has a flat landscape (i.e. the $\lambda_{\max}$ of Hessian is bounded by $\frac{2}{\eta}$). In this way, our results actually apply to all flat points found along the learning process of GD and the ''Edge of Stability'' regime. We will make this more general implication of our result clearer in the next version.
>
> **This paper is limited to univariate functions, which is not important in practice. The authors claims that the technique can be generated to multivariate functions but they did not show that.**
>
> Since this is the first paper to consider minima stability without the assumption of interpolation, we start with the univariate case. The generalization to multivariate functions includes deriving a similar weighted TV(1) bound (with $f^{\prime\prime}(x)$ replaced by the Laplacian $\Delta(f)$). Based on the analysis for the multivariate case with interpolation [Nacson et al 2022], the additional term about approximation error needs to be handled as in this submission. With a TV(1) bound, the learned function belongs to a Radon TV class, and the remaining analysis is based on the metric entropy of such a function class. We believe this to be an interesting and promising future direction.
>
> **How large is the family of target functions satisfying the BV(1) condition?**
>
> The BV(1) function class includes a wide range of well-known functions, including the more familiar first-order Holder class functions (functions with Lipschitz derivatives) and Sobolev class functions (functions where the second derivatives are square-integrable), but also cover more spatial heterogeneous functions such as linear splines with a small number of knots. For a few examples, see Fig. 2, Fig. 3 and Fig 4 of (Mammen and Van De Geer, 1997)
>
> It is a natural function class that ReLU neural networks represent. It is also a well-studied function class in the non-parameteric regression literature (e.g., Mammen and Van De Geer, 1997;  Donoho and Johnstone, 1998).
>
> Enno Mammen and Sara van de Geer, Locally adaptive regression splines.
>
> David L. Donoho and Iain M. Johnstone, Minimax estimation via wavelet shrinkage.
>
> Thanks again for the high-quality review. We hope our response could address your main concerns and we are happy to answer any further questions.

---

> > ### Author Response · Authors · 2024-08-11
> > **Any further questions?**
> >
> > Thanks again for your time in reviewing our paper! We have addressed your technical questions above and shared some perspectives.  Could you kindly let us know if our rebuttal satisfactorily resolved your concerns?
> >
> > We would love an opportunity to address any further questions and comments you may have before the author discussion period expires.

---

> > ### Comment · Reviewer_4Yyh · 2024-08-13
> >
> > I thank the authors for their rebuttal and I will keep my score.

---

> > > ### Author Response · Authors · 2024-08-14
> > > **Thank you**
> > >
> > > Thanks for acknowledging our response and for your continued support!

---

### Official Review · Reviewer_qFzY · 2024-07-16

**Soundness:** 4
**Presentation:** 4
**Contribution:** 3
**Rating:** 6
**Confidence:** 3

**Summary:**

The paper studies uni-variate regression with two layer networks and builds on the following observation: If the basin (derived form a quadratic approximation) around a given minimum is too narrow, gradient descent with a fixed step size $\eta$ will escape it, only sufficiently wide basins can capture the iterates of GD with large step sizes. It is thus reasonable to restrict the function space attainable by GD to the set of functions that can be expressed by a minimum with a wide basin. By extending a previous result to the noisy-label setting, they show that the width of the basin (or the hessian of the loss) relates to a (weighted) measure of the TV norm of the function encoded by the minimum. The function space of interest thus becomes a space functions with low TV norm. By computing covering numbers of this space, uniform generalization bounds are derived for this restricted space of functions.

**Strengths:**

The paper is very well written. The proofs in the appendix section are also well written and easy to follow.  The minima studied are non-interpolating, unlike the minima studied by a large portion of prior literature (although this work is in the univariate setting). Moreover, although the fact that large stepsizes bias towards solutions with fewer knots appears in prior work, this paper explores this interesting fact further in the noisy-label setting. The paper provides some avenues to understanding generalization when over-fitting is not benign.

**Weaknesses:**

- *The "optimized" assumption*: This assumption which effectively assumes that there exists wide and low minima is somewhat justified in lines 267-270, but I believe it is reasonable to disagree with the statement that it is mild. The experiments provided as evidence for the assumption are conducted with really large noise levels sigmas that are of the same order as $f$. The authors empirical argument could be strengthened by considering noise levels that do not dwarf the signal. It seems difficult to believe that a smooth $f$ can attain a training loss that is lower than $f_0$ when no assumption on the smoothness of $f_0$ is made. The authors should clarify why a condition on $f_0$ is not necessary for this assumption to be mild.

- The refined bounds with underparametrization: The bounds are refined by considering under parametrized settings with the width $k$ being smaller than the number of data points but the authors do not discuss its impact on the "optimized" assumption. Why would an underparametrized network be able to attain train losses smaller than a ground truth on which no assumptions are made ? I believe there are trade-offs between $\eta$, the level of underparametrization and the attained train loss that are not mentioned sufficiently by the authors.

- A final minor weakness: The work extends [Mulayoff et al 2021] in limited ways but does not exploit the added generalization to derive new conclusions. That stepsize affects the number of knots was explored before. There is now a sigma appearing in the smoothness bounds and an MSE term, yet there are no comments on what is added by their extension. The interplay between sigma and eta is not discussed, for instance, does having a large eta  but high noise means that stable functions need no have a low TV norm or is the upper bound loose?  I believe such discussions could strengthen their work as some results, namely Thm 4.3, are straightforward applications of generalization bounds for  spaces with known metric entropy.

**Questions:**

The weight function $g$ is inherited from [Mulayoff et al 2021]  but would really benefit from some more clarifications, even for completeness. It is clear that the interval $\mathcal{I}$ must be introduced to remove the weighing and allow the authors to use covering numbers for bounded TV norm spaces. Could the authors explain why it arises in theorem 4.1 ?

**Limitations:**

The limitations are discussed.

---

> ### Author Rebuttal · Authors · 2024-08-06
>
> We appreciate your high quality review and the positive score. Below we will reply to your comments.
>
> **Regarding the ''optimized'' assumption.**
>
> First of all, we politely point out that we actually made some assumptions on the smoothness of $f_0$. In line 312 of Theorem 4.4, we assume an upper bound on the TV(1) norm of $f_0$ (dependent on the learning rate $\eta$), which further implies that $f_0$ is inside the possible output function class learned by GD. Therefore, the ''optimized'' assumption can be satisfied if GD finds some smooth function that is near-optimal (in training loss) inside the possible output function class. Lastly, we will conduct experiments with smaller noise scale to check the assumption.
>
> **Regarding the refined bounds with underparametrization.**
>
> This is a very good point. Our generalization bound holds as well as the learned function is ''optimized'', while the result can be improved if a refined MSE bound could be derived. The case of underparametrization is therefore mentioned as an example for a refined MSE. In this case, the ''optimized'' assumption is not guaranteed, and we agree that it is harder for an underparameterized NN to perform better than the ground truth (compared to the overparameterized case). The relationship between whether the ''optimized'' assumption is satisfied and the number of neurons in the network is a very interesting problem, and we will think about experiments to showcase the relationship.
>
> **Regarding the interplay between $\sigma$ and $\eta$.**
>
> Thanks for the comment mentioning the interplay between $\sigma$ and $\eta$. The current bound scales as $O(\frac{1}{\eta}+\sigma)$, which means that either a small $\eta$ or a large $\sigma$ would lead to a larger TV(1) bound. The empirical evidence of such argument would require trying out different noise scales (similar to the first weakness you mentioned), and we leave these to the next version of the draft.
>
> **Regarding the weight function $g$.**
>
> The choice of $g$ is mainly due to technical reasons.  Based on inequality (41), the key idea is to bound the TV(1) norm of the output function by $\lambda_{\max}(\frac{1}{n} \sum_{i=1}^n \nabla_i \nabla_i^T)$ (the term ($\star$)). Therefore, the $g$ function is generated in the middle-step inequalities to lower bound the $\lambda_{\max}$ term by $\int |f^{\prime\prime}(x)|g(x)dx$. For more proof details, please refer to our Lemma F.1 or Lemma 4 of [Mulayoff et al 2021]. Lastly, some properties of $g(x)$ and its implication on the learned function $f$ can be found in Appendix B.
>
> Thanks again for the high-quality review. We hope our response could address your main concerns and we are happy to answer any further questions.

---

> > ### Author Response · Authors · 2024-08-11
> > **Any follow-up questions before discussion period expires?**
> >
> > Thanks again for your time in reviewing our paper! We have addressed your technical questions above.  Could you kindly let us know if our rebuttal satisfactorily resolved your concerns?
> >
> > We would love an opportunity to address any further questions and comments you may have before the author discussion period expires.

---

> > > ### Comment · Reviewer_qFzY · 2024-08-12
> > >
> > > Thank you for your clarifications.
> > >
> > > *The optimized assumption*: In Corollary 4.2, the optimized assumption, as far as I understand, is just assumed to be attainable without $f_0$ belonging to a BV space. My issue is that this assumption is unrealistic without constraints on $f_0$ and I believe it reasonable to disagree with your paragraph 267.
> > >
> > > I will restate my concern in other words: there is, in my understanding, a problem of having an unconstrained $\eta$: your result states that for a given $\eta$, \emph{if the ground truth is sufficiently$\frac{1}{\eta}$ smooth, hence making the optimized assumption achievable, then Corollary 4.2 holds. In your result $\eta$ needs to have an upperbound set by the smoothness of $f_0$ otherwise, the optimized assumption is an act of faith. This need for an upperbound contrasts with the spirit of the paper which aims to analyze "large step-sizes". I would appreciate if the authors could clarify why they believe the optimized assumption in Corollary 4.2 is mild. From my perspective, the optimized assumption is implicitly disallowing large stepsizes.
> > >
> > > *The interplays between $\eta$ and $\sigma$* (and underparametrization discussions):
> > > I believe it is very essential to include these discussions and experiments. The extension from [Mulayoff et al 2021] would be too limited otherwise.

---

> ### Author Response · Authors · 2024-08-12
> **Thanks a lot! We understand what you mean now!**
>
> Thanks for the follow-up questions!
> > In Corollary 4.2, the optimized assumption, as far as I understand, is just assumed to be attainable without $f_0$ belonging to a BV space. My issue is that this assumption is unrealistic without constraints on $f_0$ and I believe it reasonable to disagree with your paragraph 267.
>
> I see!  You are right that Corollary 4.2 is a valid statement without assuming regularity on f_0.  In fact, for Corollary 4.2 to be valid, all we need is MSE = $\tilde{O}(\sigma^2)$. The "optimized" condition was only used for convenience because we can bound $$\sqrt{MSE} = \\|f - f_0\\| \\leq \\|f - y\\| + \\|y - f_0\\| \leq 2\sigma.$$
>
> We can instead just use $$\sqrt{MSE} \leq \sigma + \sqrt{\text{TrainingLoss}}$$ so Corollary 4.2 does not need the "optimized" assumption.  TrainingLoss being a constant is relatively mild, because all 0 initialization already have a training loss of $\frac{1}{2n}\sum_i y_i^2 = O(1)$ if label $y_i$ are all bounded. If gradient descent does not diverge, then it should find solutions with lower loss than initialization.
>
> It is indeed unnatural to assume "optimized" for Corollary 4.2 and the discussion about it there is out of place.  We propose to defer that to Section 4.3 where it is actually used --- for solving a nonparametric regression task under the assumptions of the regularity of f_0.
>
> > The interplays between $\sigma$ and $\eta$ (and underparametrization discussions): I believe it is very essential to include these discussions and experiments.
>
> Good call. We can add these experiments and discussions.   We focused on discussing the regime when $\sigma$ is a constant because such a low signal-to-noise ratio setting is the conventional setting for nonparametric regression.
>
> Our theorems are not asymptotic and they work for all $\eta>0, \sigma>0$. One more technical result we can explicitly state about the interplay between $\sigma$ and $\eta$  is to expose $\sigma$ as a parameter in Theorem 3.2.  The updated statement will say that interpolation is not possible unless $\eta < 1/(\sigma n^2)$.

---

### Official Review · Reviewer_2Lhx · 2024-07-26

**Soundness:** 3
**Presentation:** 3
**Contribution:** 3
**Rating:** 7
**Confidence:** 4

**Summary:**

This paper studies the generalizability of stable local minima in univariate regression with shallow ReLU networks. Along the way, the authors provide a bound on a weighted total variation norm of networks corresponding to stable solutions which in turn provide a tighter generalization bound.

**Strengths:**

The paper is well written and provides a novel way to obtain generalization bounds using minima stability theory.

**Weaknesses:**

It is not clear to me why the experiments (e.g. figure 3) corroborate the theoretical results in the paper.

To illustrate my point, suppose for a given $\eta > 0$, a solution $\theta^*$ is linearly stable.
Now suppose one trains a ReLU NN with a different learning rate $\alpha > 0$ and it converges to $\theta^*$. When considering the generalization gap of the solution $\theta^*$ (e.g. theorem 4.4), only the constant $\eta$ is relevant. The point is that the actual learning rate of GD that was used to find stable solutions is irrelevant to the generalizability of the solution (at least in the context of theorem 4.4) . Hence, I feel that more compelling experiments would study generalization errors of stable local minima that are stable for varying $\eta$.

In this vein, I feel that the statement, “Meanwhile, the dependence $\frac 1 {\eta}$ ...” (line 296 -298) to be inexact as it is only supported by the experiments.

**Questions:**

In the context of theorem 4.3 and 4.4, should $n_I$ be the number of data points belonging to the interval $I \subset [-x_{max}, x_{max}|$ instead of the length of the interval $|I|$? Otherwise, the RHS of the bound in theorem 4.4 seems to be non-vanishing as $x_{max} $ is at least on the order of the length of the interval $|I|$.

Can the authors also provide some intuition on how the weight function $g$ was chosen?

**Limitations:**

The authors have adequately addressed the limitations.

---

> ### Author Rebuttal · Authors · 2024-08-06
>
> We appreciate your high quality review and the positive score. Below we will reply to your comments.
>
> **It is not clear to me why the experiments (e.g. figure 3) corroborate the theoretical results in the paper.  ... The point is that the actual learning rate of GD that was used to find stable solutions is irrelevant to the generalizability of the solution (at least in the context of theorem 4.4).**
>
> The reviewer is right that our generalization bounds work for all ``flat'' solutions when the sharpness is measured in terms of the $\lambda_{\max}$ of the Hessian, and it may appear to be slightly indirect to expose only the learning rate $\eta$ in the theorems.
>
> The sharpness and learning rate $\eta$ is connected in two ways. First, from the linear stability-theory, gradient descent with learning rate $>\eta$ cannot stably converge to solutions with $\lambda_{\max}$ larger than $2/\eta$, thus ruling out those solutions at the steady-state of GD dynamics. Second, it has been observed that gradient descent training of neural networks finds solutions on the edge-of-stability.
>
> To say it differently, the choice of learning rate $\eta$ controls the maximum ``sharpness'' of the solutions that GD converges to --- an implicit constraint that helps with generalization. Figure 3 clearly demonstrates this effect. Also, see the middle panel of Figure 2. We believe these results do corroborate our theoretical findings.
>
> In the example you gave, let $\theta^*$ be a solution that GD with learning rate $\eta$ converges to.  It is true that GD with an even smaller learning rate $\alpha < \eta$ can possibly converge to $\theta^*$ too, there are many other lower-loss solutions that GD with the smaller learning rate $\alpha$ can find more easily, thus do not converge to $\theta^*$.
>
> Note that even if GD with learning rate $\alpha$ does converge to $\theta^*$, it does not invalidate our theorem (replacing $\eta$ with $\alpha$ makes the bound more relaxed).
>
> **In the context of theorem 4.3 and 4.4, should $n_I$
> be the number of data points belonging to the interval $I\subset[-x_{\max},x_{\max}]$ instead of the length of the interval $|I|$?**
>
> Yes, you are correct. This is a typo we found after the submission of the draft. Indeed, $n_I$ should be the number of data points such that $x_i\in I$. Therefore, the RHS is vanishing for a fixed $x_{\max}$ and increasing number of data points. Thanks for pointing this out, we will correct this in the next version.
>
> **Can the authors also provide some intuition on how the weight function $g$ was chosen?**
>
> The weight function $g(x)$ is inherited from [Mulayoff et al 2021] and its choice is mainly due to technical reasons.  Based on inequality (41), the key idea is to bound the TV(1) norm of the output function by $\lambda_{\max}(\frac{1}{n} \sum_{i=1}^n \nabla_i \nabla_i^T)$ (the term ($\star$)). Therefore, the $g$ function is generated in the middle-step inequalities to lower bound the $\lambda_{\max}$ term by $\int |f^{\prime\prime}(x)|g(x)dx$. For more proof details, please refer to our Lemma F.1 or Lemma 4 of [Mulayoff et al 2021].
>
> Note that this is not an artifact of the proof. The weighting function $g$ correctly describes the implicit bias due to minima stability. The implicit smoothness regularity near the boundary of the distribution support is weaker than that in the center.
>
> Thanks again for the high-quality review. We hope our response could address your main concerns and we are happy to answer any further questions.

---

> > ### Author Response · Authors · 2024-08-11
> > **Any further questions / comments?**
> >
> > Thanks again for your time in reviewing our paper! We have addressed your technical questions above.  Could you kindly let us know if our rebuttal satisfactorily resolved your concerns?
> >
> > We would love an opportunity to address any further questions and comments you may have before the author discussion period expires.

---

> > > ### Comment · Reviewer_2Lhx · 2024-08-12
> > >
> > > Thank you for the detailed response, and I have raised the score accordingly.

---

> > > > ### Author Response · Authors · 2024-08-12
> > > >
> > > > Thanks again for your high-quality review and your support.

---

### Decision · Program_Chairs · 2024-09-25

**Decision:**

Accept (spotlight)

**Comment:**

This paper provides new optimization and generalization results in an interesting univariate setting.  Reviewers are uniformly positive and this is an easy accept :)